# Light Differentiable Logic Gate Networks

**Lukas Rüttgers, Till Aczel**, **Andreas Plesner\* & Roger Wattenhofer**
ETH Zürich
Zürich, Switzerland
`lruettgers,taczel,aplesner,wattenhofer@ethz.ch`

## Abstract

Differentiable logic gate networks (DLGNs) exhibit extraordinary efficiency at inference while sustaining competitive accuracy. But vanishing gradients, discretization errors, and high training cost impede scaling these networks. Even with dedicated parameter initialization schemes from subsequent works, increasing depth still harms accuracy. We show that the root cause of these issues lies in the underlying parametrization of logic gate neurons themselves. To overcome this issue, we propose a reparametrization that also shrinks the parameter size logarithmically in the number of inputs per gate. For binary inputs, this already reduces the model size by 4x, speeds up the backward pass by up to 1.86x, and converges in 8.5x fewer training steps. On top of that, we show that the accuracy on CIFAR-100 remains stable and sometimes superior to the original parametrization. *Equal contribution.

## 1 Introduction

Contemporary large, overparametrized neural networks have demonstrated remarkable expressivity (Allen-Zhu et al., 2019), but their computational cost necessitates efficiency improvements while sustaining their approximation accuracy (Gusak et al., 2022). With that goal, several approaches directly draw from the physical structure of the underlying hardware to parametrise model classes (Wang et al., 2020; Benamira et al., 2024; Bacellar et al., 2024; Hubara et al., 2016). Among them, differentiable logic gate networks maintain an unparalleled performance-efficiency trade-off (Petersen et al., 2022). Subsequent works have since advanced this model to convolutional or recurrent architectures (Petersen et al., 2024; Bührer et al., 2025). Yet, several issues like vanishing gradients, discretization errors, and high training cost impede scaling these models in depth.

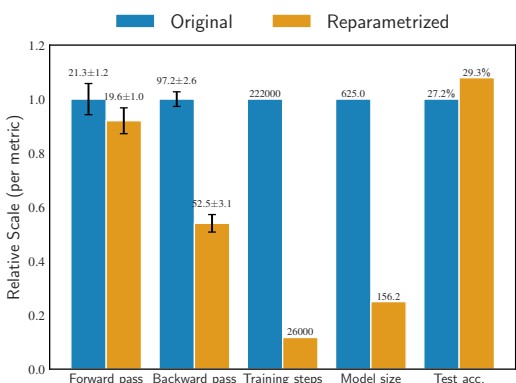

Figure 1: For a CIFAR DLGN (Petersen et al., 2022), our reparametrized DLGNs require 4x less memory, converge in 8.5x fewer training steps, and perform the forward and backward passes in up to 8% and 45% less time, respectively. Details in Section 5 and Section B.5.

So far, prior works have mainly patched these problems with alternative parameter initialization schemes (Petersen et al., 2024; Yousefi et al., 2025). But these remedies do not fully resolve the issues, as they neglect that the primary root cause lies in the parametrization of logic gate neurons themselves. For that reason, scaling the convolutional DLGN from Petersen et al. (2024) in depth still grossly degrades its discretized accuracy (cf. Figure 5b).

In this work, we tackle the DLGN parameterization, study how it gives rise to the problems mentioned above, and propose a reparametrization that overcomes the problems; the reparametrization is illustrated in Figure 2. Over and above, we explicate the impact that

initializations have on gradient stability and optimization dynamics in deep logic gate networks. In particular, we identify RIs as proposed by Petersen et al. (2024) as one of the simplest instances of a larger class of negation-asymmetric heavy-tail initializations, and elucidate why such initialization schemes are particularly beneficial for the information flow in both the forward and backward pass during training. Combining such initializations with the reparametrization, we overcome the issues mentioned above and obtain logic gate networks that are more expressive, more scalable, and more efficient to train (cf. Figure 1).

Petersen et al. (2022) showed that DLGNs can process one million MNIST or CIFAR-10 images per second on a single CPU core, Petersen et al. (2024) later showed that convolutional DLGNs take less than 10 nanoseconds per CIFAR-10 image on an FPGA, and Bührer et al. (2025) showed that recurrent DLGNs require 20'000 times fewer logic operations to deliver performance comparable to RNN, GRU, and Transformer-based models in the WMT'14 German to English translation task (Bojar et al., 2014b). These values show that DLGNs are very suitable for real-world deployment once the accuracy matches state-of-the-art models. To facilitate the research needed to close this accuracy gap, our reparametrization makes training more efficient without altering the inference dynamics that make DLGNs attractive. We find that models require 4x less VRAM to train, process backward passes up to 1.86x faster, and 8.5x fewer training steps.

## 2 Background on Logic Gate Networks & Related Work

### 2.1 Logic Gate Networks

In essence, differentiable logic gate networks differ from feed-forward neural networks in the parametrization of each neuron. In standard architectures, each neuron is a composition of vector-algebraic operations with non-linear activation functions (Fukushima, 1980; Schmidhuber, 2015; LeCun et al., 2015; Goodfellow et al., 2016). By contrast, differentiable logic gate networks (DLGNs) associate each neuron with a binary Boolean function $G : \{0, 1\}^2 \to \{0, 1\}$ (Petersen et al., 2022). That way, each neuron is connected to only two neurons in the previous layer. Combined with bit-level operations, this extreme sparsity renders DLGNs particularly suitable for high-performance inference on devices with low computational resources.

Adhering to the canonical ordering of Boolean functions (cf. Table 5), we denote the 16 binary Boolean functions by $G_i, 1 \leq i \leq 16$. We categorize these functions based on the number of non-zero outputs into four ANDs, four ORs, two constants, two XORs, and four pass-throughs, which merely forward one of the inputs, negated or non-negated. A layer of such neurons is referred to as the logic layer.

Naturally, the space of Boolean functions and the functions themselves are discrete, and thus do not immediately give rise to differentiable neurons. To apply gradient-based optimization methods, the original paper proposed to continuously relax each neuron to the probability simplex over all 16 functions (Petersen et al., 2022),

$$g(p, q) := \sum_{i=1}^{16} \omega_i g_i(p, q), \quad p, q, \in [0, 1], \quad \omega_i \geq 0, \sum_j \omega_j = 1. \tag{1}$$

where each function $g_i$ is a probabilistic surrogate of the deterministic $G_i$, defined as

$$g_i(p, q) := \mathbb{E}_{\substack{A \sim \text{Ber}(p), \\ B \sim \text{Ber}(q)}} [G_i(A, B)], \quad p, q \in [0, 1]. \tag{2}$$

Such a surrogate is necessary to deal with real-valued inputs $p, q$ during training, for which the underlying $G_i$ are not defined. Accordingly, we will refer to $\omega_i$ as the weight of $g_i$.

Moving back to the parameters of each neuron, the authors decided to initialize the weights in Equation (1) via a softmax of i.i.d. random variables

$$\omega_i = \frac{\exp(\Omega_i)}{\sum_j \exp(\Omega_j)}, \quad \Omega_i \stackrel{i.i.d.}{\sim} \mathcal{N}(0, \sigma^2). \tag{3}$$

Likewise, a softmax operation is used to eventually obtain differentiable class scores from this network for classification tasks. In particular, for $C$ classes, a layer coined GroupSum partitions the gate outputs of the final logic layer into $C$ contiguous bins and accumulates them to obtain the corresponding logits.

At inference, all these softmax operations are replaced by argmax operations. This rounds each neuron to the binary gate $g_i$ with the highest weight $\omega_i$, which yields a logic gate circuit that can be directly embedded in hardware such as FPGAs or ASICs (Zia et al., 2012). Naturally, this rounding operation entails a discretization error that might further reduce performance at deployment. We hence refer to both versions of the network as the continuous and discretized DLGN.

Contending with both this discretization error and vanishing gradients, Petersen et al. (2024) observed superior performance when they replaced the Gaussian initialization in Equation (3) by an RI, which deterministically assigns a high initial weight to the pass-through gate function $G_4(A, B) = A$,

$$\Omega_i = \begin{cases} 5, & i = 4 \\ 0, & i \neq 4 \end{cases}, i = 1, \ldots, 16.$$ (4)

Similar to the original idea of residual connections (He et al., 2016), this pass-through bias stabilized training. On top of that, it notably reduces the number of non-trivial logic gates that remain after the discretized DLGN undergoes a logic simplification. That way, they obtained a logic gate circuit that achieves a test accuracy of 85% on CIFAR-10 with less than 29 million gates, which is far less than what competitive networks required (Petersen et al., 2024, Sec. 5.1).

Albeit effective, this initialization is still subject to limitations that arise from the underlying parametrization, which we will pinpoint in Section 3. But first, we present other related work and explain how they differ from DLGNs in both their reparametrized and original form.

## 2.2 Other Related Work

Several works have exploited that learning circuits of logic gates with more than two inputs allows for embedding more functional expressivity on the same hardware (Umuroglu et al., 2020; Bacellar et al., 2024). On the contrary, DLGNs were practically limited to learn logic gates with very few inputs, as processing $2^{2^n}$ parameters per logic gate with $n$ inputs quickly becomes intractable. With our reparametrization that reduces the number of parameters to $2^n$, advancing DLGNs to process more than two inputs per gate becomes a viable option.

In contrast to our reparametrization, these works do not directly estimate the outputs of the logic gates. Instead, they use a different representation class and quantize this class to logic gates after training. However, these indirect representations either fall short of exploiting the expressivity of logic gates (Umuroglu et al., 2020) or are costlier to parametrize (Andronic & Constantinides, 2023; 2025). We provide a detailed comparison in Section H.

## 3 Reparametrizing Logic Gate Neurons

### 3.1 Weaknesses of the current parametrization

We demonstrate that redundancies in the parametrization are the primary cause of vanishing gradients and large discretization errors.

#### 3.1.1 Gradient stability

Each Boolean function $G_i$ has a negated counterpart. Adhering to the canonical ordering of Boolean functions (cf. Table 5), we denote this counterpart by $G_{\neg i} := G_{17-i} \equiv \mathbf{1} - G_i \equiv \neg G_i$. The same holds for the probabilistic surrogates $g_i$. Under this condition, choosing independent weights for each $g_i$ and its negated counterpart $g_{\neg i}$ is fatal, as it provokes self-

cancellations in the partial derivatives, progressively diminishing the gradient norm during backpropagation.

To see this, we equally denote $\omega_{\neg i} := \omega_{17-i}$ to expose the symmetry in Equation (1) as

$$g(p,q) := \sum_{i=1}^{8} \omega_i g_i(p,q) + \sum_{i=1}^{8} \omega_{\neg i} g_{\neg i}(p,q) \tag{5}$$

$$= \sum_{i=1}^{8} \omega_i g_i(p,q) + \omega_{\neg i}(1 - g_i(p,q)). \tag{6}$$

Having i.i.d. $\omega_i$, this translates to a weighted sum of sign-symmetric random variables in the partial derivatives

$$\frac{\partial g(p,q)}{\partial p} = \sum_{i=1}^{8} (\omega_i - \omega_{\neg i}) \frac{\partial g_i(p,q)}{\partial p}. \tag{7}$$

Initializing $\Omega_i$ with the default variance $\sigma = 1.0$ will concentrate the gradient norm around 0 (cf. Figure 23a) and entail vanishing gradients with high probability, as Petersen et al. (2024) have already encountered. Notably, even with a variance as large as $\sigma^2 = 16.0$, many partial derivatives remain concentrated at zero (cf. Figure 23b).

RIs as proposed by Petersen et al. (2024) successfully break this sign-symmetry to

$$\frac{\partial g(p,q)}{\partial p} = \sum_{i=1}^{8} (\omega_i - \omega_{\neg i}) \frac{\partial g_i(p,q)}{\partial p} \stackrel{(3)}{=} \sum_{i=1}^{8} \frac{e^{\Omega_i} - e^{\Omega_{\neg i}}}{\sum_j e^{\Omega_j}} \frac{\partial g_i(p,q)}{\partial p} \stackrel{(4)}{=} \frac{e^{\Omega_4} - 1}{e^{\Omega_4} + 15}. \tag{8}$$

Once more, symmetric overparametrization traps RIs in a tension between maintaining gradient stability and stalling optimization for other gate functions (cf. Section F.1).

While sign-symmetries interfere with the gradient signal in a destructive way, there are also other redundancies in the parametrization that contribute to the discretization error.

### 3.1.2 DISCRETIZATION ERROR

When converting the continuous relaxation to a logic gate circuit, the softmax-to-argmax rounding principle (cf. Section 2.1) discretizes each neuron to the logic gate function with the highest weight $\omega_i$. But with the redundancies in this parametrization, the logic gate that is rounded to is not necessarily the one that the neuron is closest to. For example, assume a neuron with weight 0.4 for the one pass-through gate $G_4(A, B) = A$, weight 0.3 for the other pass-through gate $G_6(A, B) = B$, and weight 0.3 for the OR function $G_8(A, B) = A \vee B$. For the four binary inputs $(0, 0), (0, 1), (1, 0), (1, 1)$, the neuron will output $0, 0.6, 0.7, 1$. Clearly, this output behaviour is closest to the OR function $G_8$, although the argmax is the pass-through gate $G_4$. Argmax is effective only when applied to inputs that are exclusive and independent.

Redundancies in the parametrization are the leading cause of vanishing gradients. In the following, we present an exact, redundancy-free parametrization.

### 3.2 INPUT-WISE PARAMETRIZATION

In fact, each binary function $G : \{0,1\}^2 \to \{0,1\}$ has a unique decomposition

$$G(k,\ell) = \alpha_{00} E_{00}(k,\ell) + \alpha_{01} E_{01}(k,\ell) + \alpha_{10} E_{10}(k,\ell) + \alpha_{11} E_{11}(k,\ell), \tag{9}$$

where $\alpha_{ij} \in \{0, 1\}$, and $E_{ij}$ is the indicator function $E_{ij}(k,\ell) = \mathbb{1}\{(k,\ell) = (i,j)\}$. This exact representability also transfers to the probabilistic surrogates

$$g = \alpha_{00} e_{00} + \alpha_{01} e_{01} + \alpha_{10} e_{10} + \alpha_{11} e_{11}, \tag{10}$$

where $e_{ij}(p,q) = \mathbb{E}[E_{ij}(p,q)]$ as in Equation (2). Relaxing the binary coefficients $\alpha_{ij}$ to the continuous interval $\omega_{ij} \in [0,1]$ and rounding back via $\omega_{ij} > 0.5$, we obtain the exact parametrization

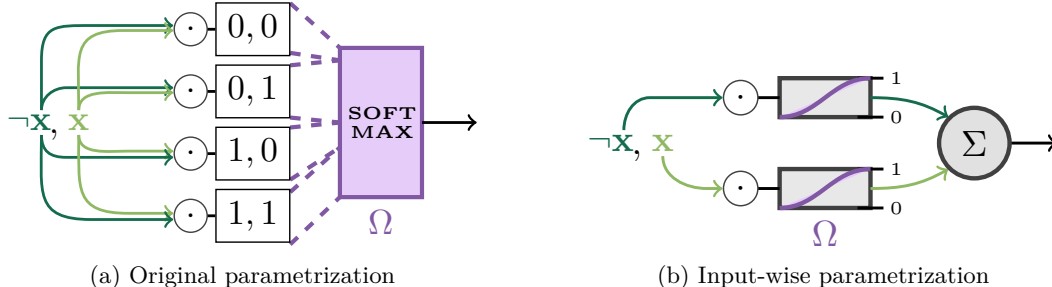

(a) Original parametrization        (b) Input-wise parametrization

Figure 2: Illustrating the reparametrization for logic gates with one input. It requires only $2^n$ learnable parameters $\Omega$ for $n$ inputs, opposed to $2^{2^n}$ for the original parametrization.

$$
\begin{aligned}
g_\omega(p, q) = (1-p) \cdot (1-q) \cdot \omega_{00} \\
+(1-p) \cdot q \quad\quad \cdot \omega_{01} \\
+ \quad\; p \cdot (1-q) \cdot \omega_{10} \\
+ \quad\; p \cdot q \quad\quad \cdot \omega_{11}.
\end{aligned} \tag{11}
$$

Similar to Petersen et al. (2022), one could learn such a bounded coefficient $\omega_{ij} \in [0,1]$ by mapping a real parameter $\Omega_{ij} \in \mathbb{R}$ to an activation function $\rho : \mathbb{R} \to [0,1]$. We will defer the specific function choice to Section C.1.1 and stick with the standard sigmoid function for now, i.e. $\rho(x) := \frac{1}{1+\exp(-x)}$.

Since the basis of the class of Boolean functions with $n$ inputs has cardinality $2^n$, this equally expressive parametrization requires logarithmically fewer parameters than the soft-max parametrization used by Petersen et al. (2022), which assigns an individual parameter to each of the $2^{2^n}$ Boolean functions. For the class of binary functions used here, this already shrinks the model size by a factor of 4, and renders learning higher-dimensional Boolean functions computationally more viable. We hence also refer to this reparametrization as input-wise parametrization (**IWP**), and use the abbreviation **OP** for the original parametrization.

### 3.3 No gradient stability without appropriate initializations

We now show that the IWP eliminates the pathways causing gradient cancellations and discretization errors. Any remaining gradient instability arises from other architectural factors, particularly parameter initialization.

To begin with, rounding the outputs of $g_\omega$ to their closest binary numbers clearly attains minimal errors with respect to any Minkowski norm and any other norm that is based on a uniform distance metric between outputs of the function. Proof in Section F.3.

Moving on with gradient stability, the partial derivative now becomes

$$
\frac{\partial g_\omega(p, q)}{\partial p} = (1-q)(\omega_{10} - \omega_{00}) + q(\omega_{11} - \omega_{01}) \tag{12}
$$

$$
= \mathop{\mathbb{E}}_{B \sim \mathrm{Ber}(q)} \left[ \omega_{1B} - \omega_{0B} \right]. \tag{13}
$$

An i.i.d. parameter initialization with sufficiently low variance would still entail cancellations, but, opposed to the OP, the IWP itself does not compound this problem further. Heavy-tail initializations that concentrate most weights $\omega_{ij}$ close to $0, 1$ would already resolve these cancellations inside a neuron to a sufficient extent, as we explain exhaustively in Section F.4. But a heavy tail alone is not enough in general. As long as the parameter initialization treats each function and its negated counterpart independently, gradients will distribute sign-symmetrically between different neurons during backpropagation. The more subsequent gates a neuron passes its output to, the more likely it is that the sum of partial derivatives that it receives during backpropagation will concentrate at 0. Therefore, appropriate initialization schemes should be negation-asymmetric as well.

A residual initialization (RI) as proposed by Petersen et al. (2024) that assigns a high initial bias to the pass-through $G_4(A, B) = A$ is a simple instance satisfying both requirements. More complex instances, like an AND-OR initialization that concentrates each gate either to the AND or OR function, are also feasible in principle. However, it turns out that RIs entail a gate output distribution (cf. Figure 3) that organizes the optimization of logic gate networks consecutively from earlier to later layers, which is advantageous for training deep networks. We substantiate this argument in Section F.4.1, where we study the class of heavy-tail, negation-asymmetric initialization schemes in more detail.

To conclude, we pair our IWP with RIs and show that the result is more scalable in depth and expressive complexity.

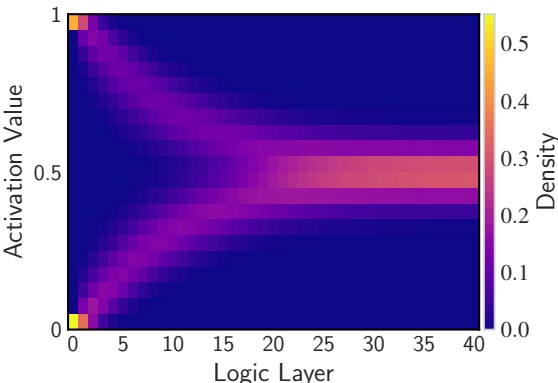

Figure 3: Distribution of gate outputs for an IWP DLGN right after residual initialization (RI), averaged over 100 images of CIFAR-100. That way, RI postpones gate learning in later layers until earlier layers are more refined. This incremental refinement allows to learn complex deep networks.

## 4 EXPERIMENTS

To verify our claims of better gradient stability, discretization accuracy, and training efficiency of our IWP, we adopt the original DLGN models and the experimental training setup from Petersen et al. (2022). We also cover the models and experimental setup from Petersen et al. (2024) to show the benefits apply to CDLGNs as well.

### 4.1 BENCHMARKS

In both works, the networks were evaluated on several image classification benchmarks, with CIFAR-10 (Krizhevsky, 2009) as the most challenging dataset. However, the shallow models used there already perfectly fit the training dataset after a few iterations, which restricts the measurability of further expressive benefits when scaling the networks in depth. Thus, we decided to lift the complexity of the task in two ways: Firstly, we transition to CIFAR-100, a 100-class extension of CIFAR-10 (Krizhevsky, 2009). Secondly, we employ random resized crops and horizontal flips as standard data augmentations (PyTorch Core Team, 2023).

We need to account for the 10-fold class increase in the final prediction head of the model. Apart from that, no further adjustments to the original experimental setup for CIFAR-10 are required. The class increase can be encountered in two different ways. Following recommendations of Petersen et al. (2024, Appendix A.2), we explore both options; see Section D.2.1 for details.

As a trade-off between computational feasibility and expressiveness, we finally decided to consider the medium-sized M models for both papers. When we refer to the DLGN and CDLGN in the experiments, we hence always mean the specific CIFAR-10 M architecture from Petersen et al. (2022, Appendix A.1) and Petersen et al. (2024, Appendix A.1.1), respectively. To estimate uncertainty, we train each model on three different seeds.

### 4.2 IMPLEMENTATION OF REPARAMETRIZATION

To implement our IWP and the adjusted initialization schemes in the given Python and CUDA implementation, we merely have to override the weight initialization and the forward and backward functionality according to Equations (11) and (12).

While we assumed the sigmoid function as the binary gate output estimator $\rho$ in Equation (11) for the sake of exposition, we observed slightly superior expressivity with a rescaled sinusoidal estimator $\sin_{01}(x) = 0.5 + 0.5\sin(x)$ and adopted that one for subsequent experiments. See Section C.1.1 for details.

### 4.3 Scaling models in depth

Eventually, we want to reliably assess how increasing model depth affects performance for both parametrizations. To scale both DLGNs and CDLGNs in a comparable, architecture-agnostic way, we introduce a depth-scale parameter $D \in \mathbb{N}$, and obtain depth-scaled networks by placing $D$ (convolutional) logic layers instead, where only one was placed in the original architecture. Section D.1 presents implementation details of this depth scaling.

## 5 Results

### 5.1 Reparametrization reduces vanishing gradients

To begin with, vanishing gradients as the major hindrance for scaling DLGNs in depth, the input-wise parametrization drastically reduces the shrinkage of gradient norm as we backpropagate over layers. As Figure 8a showcases, the gradient norm undercuts machine precision after 16 logic layers already, and vanishes to $10^{-34}$ over 40 layers, when the OP is used. But as already discussed in Section 3.3, an IWP alone without appropriate negation-asymmetric, heavy-tail initializations can not reduce the gradient norm shrinkage sufficiently and also ends up with an average gradient norm of $10^{-16}$ after 40 layers.

### 5.2 Residual initializations scale best with depth

The residual initialization (RI), although biasing towards a single gate function only, proves most effective for training deep DLGNs. On the one side, all other single-gate biases quickly concentrate the gate outputs at one value (cf. Figure 27) where their gradients become 0 and stifle gradient flow (cf. Figure 24b). On the other side, some multi-gate biases appeared competitive alternatives to RI, such as the AND-OR initialization, which exerts a theoretically more appealing anticoncentration that retains inputs close to 0 and 1 over the layers (cf. Figure 27e). Nonetheless, these methods remain slightly inferior to RI in terms of both gradient stability (cf. Figure 24a) and accuracy (cf. Figure 26a). While the former drawback is rather obvious because the pass-through gate $G_4$ is unparalleled in retaining a uniformly high gradient of 1, the latter relates to the more intricate discrepancy in the optimization dynamics that each of the two initialization schemes gives rise to. As discussed in Section F.4.2, RIs order optimization of neurons from earlier to later layers. On the contrary, AND-OR initializations allow for non-uniform updates of the four gate outputs for neurons at later layers right from the beginning. This additional freedom, however, seems not only detrimental to the accuracy of the continuous relaxation. Surprisingly, despite its anti-concentration, this alternative initialization grossly compounds to the discretization error as we further increase depth (cf. Figure 26b).

### 5.3 Reparametrization drastically strenghtens hyperparameter robustness

Hyperparameters needed to be carefully chosen for the original DLGN. Besides the choice of optimizer and learning rate, the softmax temperature $\tau$ of the GroupSum layer was already identified a highly sensitive parameter in Petersen et al. (2022). By contrast, we find that IWP DLGNs are far more robust than OP DLGNs against variations of these hyperparameters. When swapping the Adam optimizer for established alternatives such as Stochastic Gradient Descent (SGD), Nesterov Accelerated Gradient (NAG), or Adadelta, the test accuracy of the OP DLGN deteriorates drastically while the IWP DLGN maintains a stable discretized test accuracy of above 30% for all of them (cf. Figure 4a). On top of that, the large discretization errors of the OP DLGNs become particularly apparent when shrinking the GroupSum temperature $\tau$ (cf. Figure 4b). Conversely, raising the temperature already reduces the test accuracy of the continuous OP DLGN. A complete

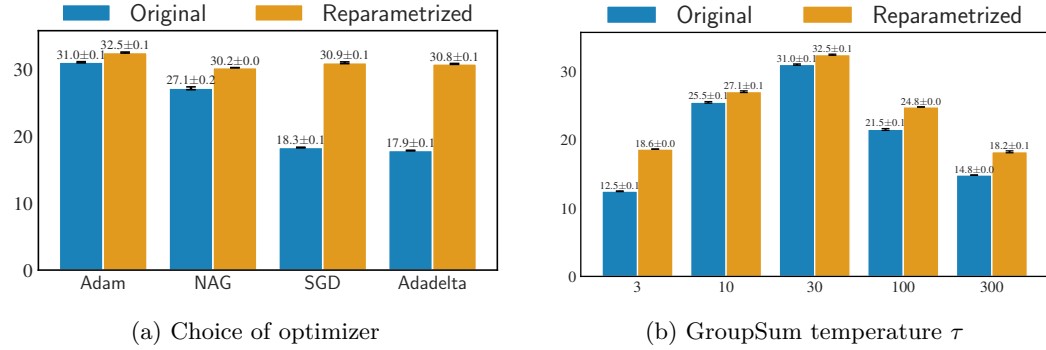

(a) Choice of optimizer

(b) GroupSum temperature $\tau$

Figure 4: Discretized test accuracy of the CIFAR DLGN with 3-fold depth when varying hyperparameters, averaged over two seeds. Details in Section B.1.

overview of continuous and discrete accuracies is provided in Table 2. We continue these stability experiments in Section D.2, where we show that IWP DLGNs remain consistently superior to the OP for different learning rates and binary input encodings aswell.

## 5.4 ORIGINAL PARAMETRIZATION SCALES WORSE WITH DEPTH

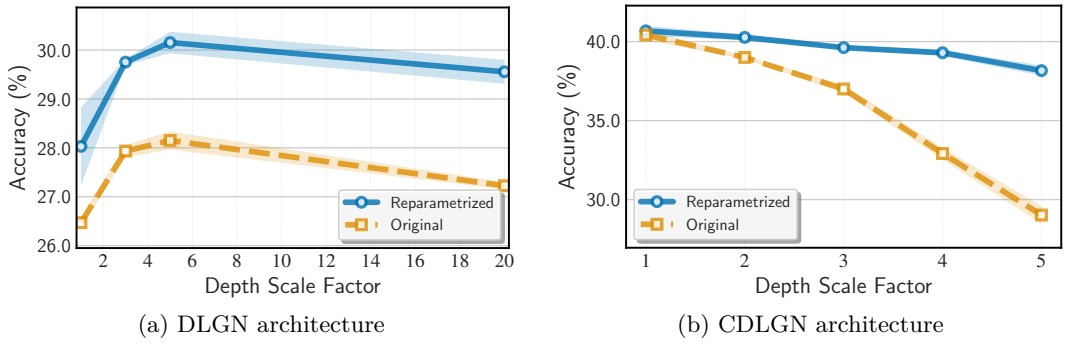

(a) DLGN architecture

(b) CDLGN architecture

Figure 5: Discretized test accuracy, averaged over three seeds, when scaling the CIFAR M DLGN (Petersen et al., 2022) and CDLGN (Petersen et al., 2024) in depth.

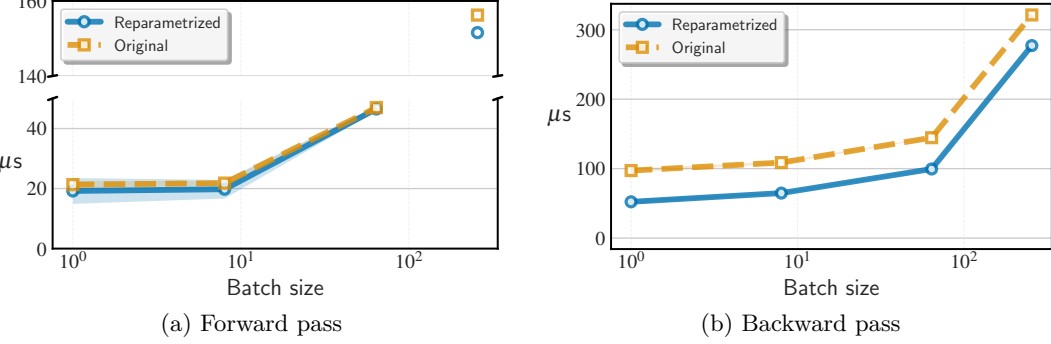

(a) Forward pass

(b) Backward pass

Figure 6: Training times for the DLGN with 20-fold depth. Mean and standard deviation were computed over 20 batches of CIFAR-100.

RIs also suppress the undesirable properties of the OP but cannot fully level them out, demonstrating that an inappropriate underlying parametrization can irreversibly condition shortcomings for optimization. The IWP addresses these weaknesses, and pairing it with an RI achieves superior performance to the OP with an RI. For one, the IWP with RI

still retains a higher gradient norm than the OP with RI (cf. Figure 8b). For another, we observe a clear gap in the predictive performance as well. For the DLGN, this gap is already apparent with baseline depth scale $D = 1$. Moreover, the IWP consistently maintains this gap, which the OP cannot close even with 20-fold depth (cf. Figure 5a) and eventually plateaus at roughly 28% test accuracy. For the CDLGN, the shallow baseline network performs almost equivalently. But increasing depth now begins to expose a drastic performance gap that culminates in a more than 1.3 times better test accuracy of the IWP for $D = 5$ (cf. Figure 5b).

We observe that this gap is mainly attributable to a large discretization error for the OP. Figure 9b shows that the accuracy of the continuous OP CDLGN trails the IWP only by a few percent. Unfortunately, the increasing depth ceases to benefit performance for the IWP as well at some point, at least when not increasing the number of optimization steps. Henceforth, we suppose that this is caused by shared underlying characteristics of the overall DLGN architecture, and discuss potential reasons later in Section 6.

## 5.5 Training Efficiency

By reducing the number of real parameters per neuron from 16 to 4, we shrink the model size by a factor of 4 (cf. Figure 1). This reduction also reduces the working set size during the forward and backward passes in the CUDA kernel. This advantage becomes particularly apparent for small batch sizes, where the parameter tensors dominate the memory footprint. For an 80-layer DLGN trained with batch size 1, we observe a 1.86x speedup for the backward pass and a 1.11x speedup for the forward pass (cf. Figure 6).

However, for large batch sizes, as they are typically used during training of such large models, the parametrization plays an increasingly negligible role in the overall memory and operation usage, and the relative speedup over the OP fades. We discuss further potential efficiency improvements in Section B.5.

Besides the models being lighter, the significant benefit of our IWP lies in the better gradient signal. In Figure 11 (Section B.5.1), we see that IWP converges in 8.5x fewer training steps than the OP, and as the steps are slightly faster, this means we can converge more than 8.5x faster in terms of wall clock time.

## 5.6 Consistent improvements across vision and language benchmarks

The superior performance of IWP DLGNs is not particular to CIFAR-100, but holds across several vision datasets (cf. Table 1). We evaluate DLGNs on five vision datasets ranging from the simple MNIST to the more challenging vision dataset ImageNet32 (Krizhevsky, 2009; LeCun et al., 2010; Russakovsky et al., 2015; Xiao et al., 2017), and on the NLP English to German translation task (Bojar et al., 2014a). For contextualization, we compare the achieved vision accuracies to a vanilla convolutional neural network (CNN) (Krizhevsky et al., 2012) with 2 convolutional layers that get as input the same quantized low-resolution inputs as the DLGNs.

| Model | ImageNet32 | CIFAR-100 | CIFAR-10 | Fashion-MNIST | MNIST | WMT'14 |
|---|---|---|---|---|---|---|
| DLGN OP | $4.84 \pm 0.02$ | $27.7 \pm 0.05$ | $55.33 \pm 0.23$ | $81.39 \pm 0.07$ | $92.43 \pm 0.17$ | $15.11 \pm 0.64$ |
| DLGN IWP | $4.93 \pm 0.02$ | $29.5 \pm 0.02$ | $57.47 \pm 0.20$ | $82.34 \pm 0.15$ | $94.02 \pm 0.08$ | $17.38 \pm 0.04$ |
| CNN | $5.19 \pm 0.37$ | $39.2 \pm 0.07$ | $64.01 \pm 0.16$ | $77.66 \pm 0.85$ | $92.91 \pm 1.61$ | $-$ |

Table 1: Discretized test accuracy (%,↑) across vision datasets for the DLGN, compared to a vanilla 2-layer CNN (Krizhevsky et al., 2012), and corpus BLEU scores (↑) for the NLP WMT'14 English to German task. For ImageNet32, we consider DLGNs with 3-fold depth. Details in Section D.4.

## 6 Discussion

IWP DLGNs are not prone to performance degradation for increasing depth, as they mitigate the discretization error and improve gradient stability. On top of that, they are more robust against variations in optimization and input distribution hyperparameters. However, scaling these networks in depth did not yield large expressivity benefits. And DLGNs still have a considerable generalization gap despite data augmentations. We want to discuss how to overcome both problems in the following, and present further avenues for future research.

### 6.1 Remaining expressivity bottlenecks in DLGNs

Although scaling DLGNs in depth provides slight benefits, the expressive advantage of deep DLGNs fades beyond a certain depth despite the IWP. This is expected, as the CDLGN baseline with depth $D = 1$ already contains 15 learnable gate layers. Reducing initialization variance does not alleviate this expressivity bottleneck (cf. Figure 10a). We hypothesize that this limitation does not arise from the expressivity of DLGNs. Instead, we identify possible bottlenecks in the randomized, fixed connection topology and input preprocessing. We expose the former bottleneck in Section B.7, where we show that DLGNs fail to leverage higher input resolution in contrast to CNNs. The latter bottleneck can be seen from Table 1, where the vanilla CNN performs comparably to DLGNs when provided with the same low-resolution quantized input data. An encoding-aware connection heuristic or even learned connections, as in Bacellar et al. (2024), might overcome these limitations.

### 6.2 Closing the generalization gap of DLGNs

Even before discretization, IWP DLGNs only slightly outperform the OP on test accuracy, despite a substantial increase in the training set (cf. Figure 9). Dataset augmentations alone do not close this gap, and standard techniques like dropout (Srivastava et al., 2014), random interventions, or residual connections (He et al., 2016) fail to improve test performance (cf. Section G). Designing constraints that promote generalizable functionality in DLGNs remains an open problem.

### 6.3 Learning gates with more inputs

As discussed in Section 2.2, advancing DLGNs to learn logic gates with more than two inputs finally becomes a viable option. We showcase several benefits for DLGNs with six-input gates, namely stronger expressivity (cf. Figure 14) and another 8.4x convergence speedup in the number of training steps (cf. Figure 15). Moreover, arity 6 DLGNs could also result in more efficient hardware embeddings on modern FPGAs that typically admit six inputs to their lookup tables (Bacellar et al., 2024; Zia et al., 2012). We leave this avenue to be explored in future research.

## 7 Conclusion

We proposed an input-wise parametrization (IWP) of logic gate networks with tailored initializations that allow scaling DLGNs in depth without degrading performance, while reducing parameter count logarithmically in the number of inputs per gate. Moreover, the IWP is significantly more robust against variations in hyperparameters and reduces the training time. We demonstrate the benefits of IWP across vision classifications (MNIST to ImageNet32), and a language translation task (WMT'14 English to German). That way, it facilitates research for learning logic gate circuits that are not only far deeper, but also far more complex per logic gate by increasing the number of gate inputs.

Finally, closing the generalization gap in DLGNs has become a pressing problem because the IWP and the higher gate arity notably increase the expressivity of DLGNs. But in view of the appealing performance-efficiency trade-off, DLGNs continue to lend themselves for deployment on computationally restricted hardware like real-time systems or edge devices, and advancing their potential remains a promising avenue for future research.

## Reproducibility

The source code of IWP DLGNs and the associated experiments is made available soon on GitHub[1]. There, a step-by-step guide explains how to set up the runtime environment in which we conducted our experiments, and how to reproduce any particular experiment in this environment. That way, we hope to make our experiment infrastructure as conveniently accessible as possible. To guarantee the reproducibility of our experiments, we restrict PyTorch to deterministic algorithms and fix the seeds of the random number generators that are used for the randomized initialization of weights and connections and for data loading.

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

| $\tau$ | IWP DLGN (cont.) | IWP DLGN (disc.) | OP DLGN (cont.) | OP DLGN (disc.) |
|---|---|---|---|---|
| 3 | $26.53 \pm 0.05$ | $18.64 \pm 0.01$ | $26.21 \pm 0.20$ | $12.48 \pm 0.05$ |
| 10 | $29.13 \pm 0.12$ | $27.05 \pm 0.12$ | $28.02 \pm 0.04$ | $25.49 \pm 0.11$ |
| 30 | $32.33 \pm 0.25$ | $32.46 \pm 0.08$ | $30.99 \pm 0.04$ | $31.02 \pm 0.09$ |
| 100 | $24.75 \pm 0.01$ | $24.81 \pm 0.02$ | $21.52 \pm 0.12$ | $21.52 \pm 0.13$ |
| 300 | $18.25 \pm 0.12$ | $18.25 \pm 0.14$ | $14.85 \pm 0.04$ | $14.84 \pm 0.03$ |

Table 2: Test accuracies (continuous and discretized) for CIFAR DLGNs with 3-fold depths when varying the GroupSum temperature $\tau$. While the continuous test accuracy of the OP DLGNs closely tracks the IWP for smaller temperatures, the large discretization error exposes a clear gap in the discretized accuracy compard to the IWP. For higher temperatures, the OP DLGNs are already worse in their native continuous version. Mean and standard deviation are evaluated on the CIFAR DLGN with 3-fold depth trained on two seeds.

## A  USAGE OF LLMS

We have used LLMs to polish the writing of this paper and for code generation through chats, Cursor, and Claude code. ChatGPT, Claude, Gemini, and Grammarly were employed for spellchecking, refining and condensing text, and reviewing to improve clarity and readability. Furthermore, ChatGPT, Claude, and Cursor were used to assist with code completion and generate visualizations. These tools served as auxiliary aids for writing and implementation, while all core research ideas, experimental design, and interpretation of results are our own.

## B  FURTHER EXPERIMENT RESULTS

### B.1  SUPERIORITY ACROSS HYPERPARAMETERS AND INPUT DISTRIBUTIONS

#### B.1.1  OPTIMIZER

All new optimizers, i.e. SGD, NAG and Adadelta, use the same learning rate of 100. Lower learning rates slowed down convergence for both the OP and IWP. Figure 7b substantiates that the IWP does not only previal on this particular learning rate, but remains superior across this range of learning rates.

#### B.1.2  GROUPSUM TEMPERATURE

To convert the real-valued inputs $x \in [0, 1]$ to binary encodings, Petersen et al. (2022) adopt the thermometer encoding $x_{th} := (x > t_1, t > t_2, \ldots, x > t_k)$, where $t_i = i/k + 1$ are evenly spaced thresholds in $[0, 1]$ (Carneiro et al., 2015). We extend our experiments beyond this particular choice and evaluate DLGNs on a diverse range of encodings, namely

1. a binary number system encoding, where 1s express 2-moduli at various granularity,

2. a graycode encoding, where 1s express how centered the value is at various subintervals, and

3. a sparse interval indicator encoding, where a 1 is placed only for the interval which the value is contained in.

For any of these encodings, the IWP attains superior performance (cf. Figure 7a). The same holds for deviating the learning rate from the default 0.01 (cf. Figure 7b).

### B.2  VANISHING GRADIENTS IN DEEP DLGNS

While the IWP eliminates cancellations inside a neuron, cancellations between partial derivatives of different neurons are out of the control of the parametrization. For that reason, IWP alone does not reduce the gradient norm shrinkage sufficiently, and also ends up with an

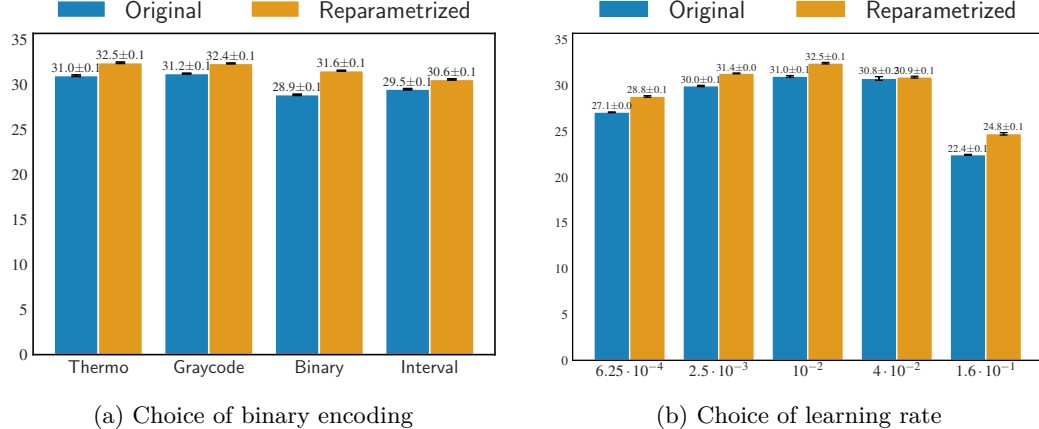

(a) Choice of binary encoding      (b) Choice of learning rate

Figure 7: IWP DLGNs achieve consistently superior test accuracy across variations of the experimental setup. Here, we compare different input encodings and deviate the learning rate from the default value $10^{-2}$ that was used in Petersen et al. (2022).

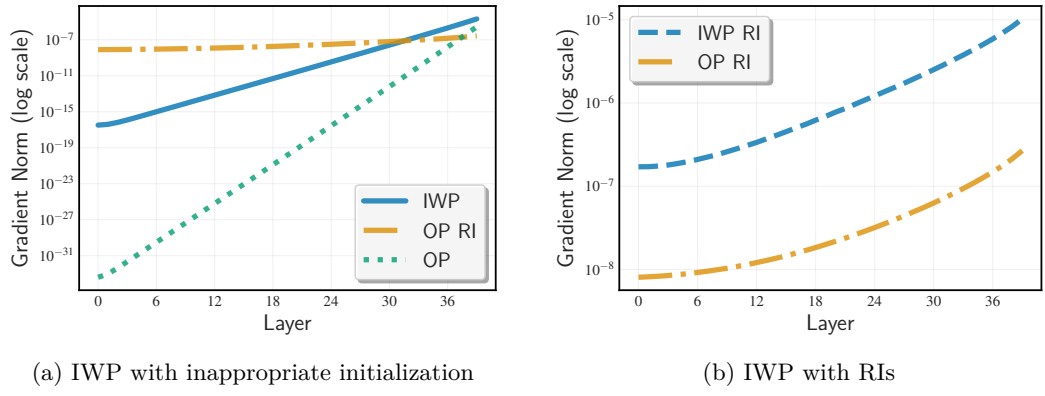

(a) IWP with inappropriate initialization      (b) IWP with RIs

Figure 8: Gradient norm decrease of an IWP DLGN with 40 layers.

average gradient norm of $10^{-16}$ after 40 layers. Avoiding this requires appropriate negation-asymmetric, heavy-tail initializations as already discussed in Section 3.3.

## B.3   Discretization error of OP

The discretization error is one major reason for the performance decrease of the OP for deeper models. For five-fold depth, the discretization gap is already substantial for both the DLGN and CDLGN architecture (cf. Figure 9).

## B.4   Deep networks do not require lower initialization variance

Deeper IWP CDLGNs with RIs do neither converge faster nor improve test accuracies (cf. Figure 10a) when lowering the initialization variance and concentrating the weights $\omega_{ij}$ closer to $0, 1$, as illustrated in Figure 10b. We believe that this is also attributable to the implicit organization of neuron optimization for RIs (cf. Section F.4.2).

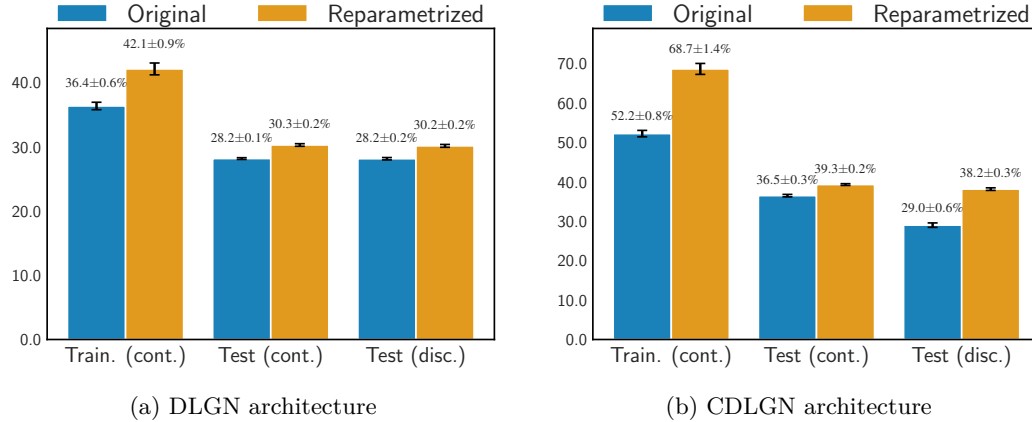

(a) DLGN architecture        (b) CDLGN architecture

Figure 9: Accuracies of the DLGN and CDLGN with five-fold depth on CIFAR-100

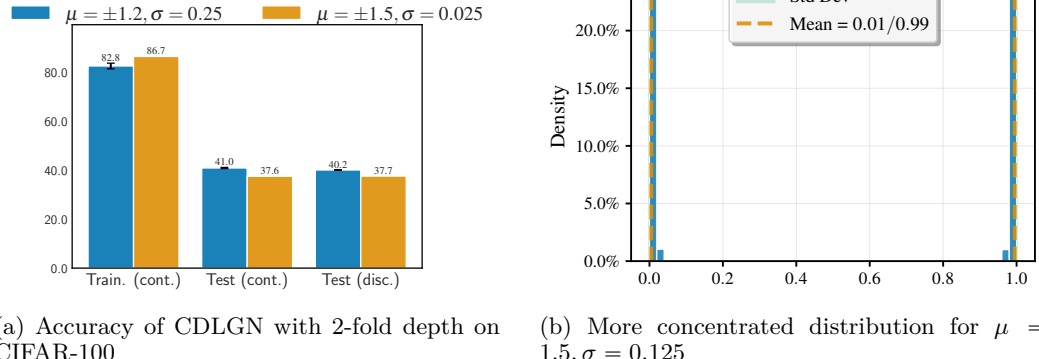

(a) Accuracy of CDLGN with 2-fold depth on CIFAR-100    (b) More concentrated distribution for $\mu = 1.5, \sigma = 0.125$

Figure 10: Reducing the initialization variance by concentrating the weights $\omega_{ij}$ even further at the tails $0, 1$ for deeper models does not improve performance.

## B.5   Training Efficiency

### B.5.1   Faster convergence of IWP

We show in Figure 11 a roofline plot (running maximum) of the test accuracy for 20-fold depth models under our IWP and the OP. We show in red the maximum accuracy of the OP, which is achieved after 222000 steps. Meanwhile, our IWP achieves this after only 26000 steps. Thus, we can converge 8.5x faster in the number of training steps. As shown in Figure 6, the steps under IWP are as fast or faster than the OP. Thus, we can also train more than 8.5x faster in terms of wall-clock time.

### B.5.2   Minimal Efficiency Impact of Gate Output Estimator

We observe that the choice of the gate output estimator $\rho(x)$ has a noticeable impact on both the runtime of the forward computation, but only a minimal effect on the backward pass. We compare the sinusoidal gate output estimator $\rho(x) = 0.5 + 0.5 \cdot \sin(x)$ with a custom double-capped linear $\rho(x) = \max(0, \min(1, x))$, whose gradient is set to 1 throughout. This not only avoids arithmetic operations during the forward and backward pass, but it also alleviates memory requirements because the constant gradient does not require saving the particular input tensor for the backward pass. Although the forward pass speeds up by 22%, the computationally dominant runtime of the backward pass reduces only by 4% (cf.

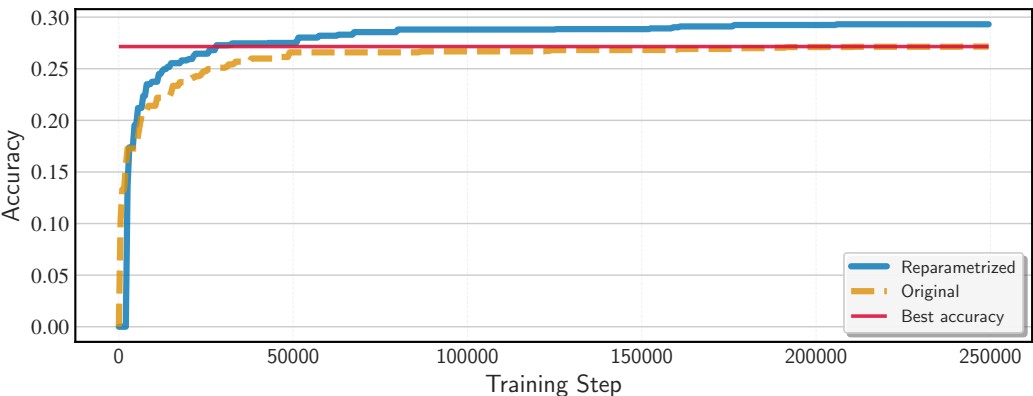

Figure 11: For the DLGN with 20-fold depth, we juxtapose the best discretized accuracy that has been achieved so far during training for both parametrizations. The OP reaches its best accuracy after 222000 steps, which is indicated by the red roofline. The IWP already surpasses this accuracy after only 26000 steps. It hence achieves more than 8.5x faster convergence.

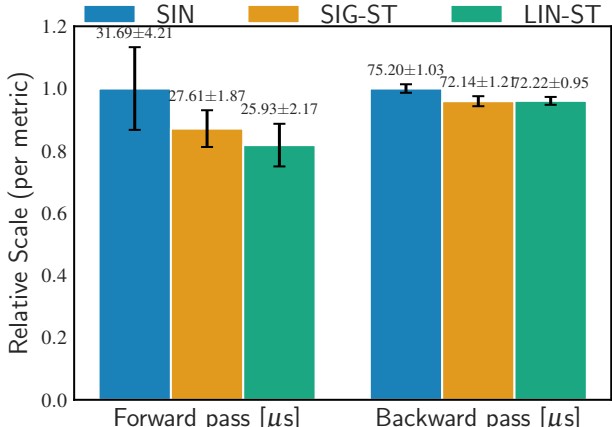

Figure 12: Runtime of the forward and backward pass for an IWP DLGN with different gate output estimators. The default sinusoidal estimator (SIN) is compared to a straight-through sigmoid (SIG-ST) and the linear straight-through estimator (LIN-ST) as introduced in Section B.5.2. Measurements are taken for a DLGN of 20-fold depth and are averaged over 20 batches with 25 CIFAR-100 instances.

Figure 12). After all, we have not measured whether a linear straight-through estimator can meet the performance of the sinusoidal estimator.

## B.6 LEARNING LOGIC GATES WITH HIGHER ARITY

The exact, redundancy-free reparametrization of logic gates finally allows to optimize circuits with gates of higher arities $n > 2$. This further leverages the expressive potential of modern FPGAs which typically admit $n = 6$ inputs to their lookup tables (Bacellar et al., 2024; Zia et al., 2012). To showcase the computational feasibility, gradient stability and optimization efficiency of training DLGNs with such an arity, we have implemented and profiled CUDA kernels for $n = 4$ and $n = 6$.

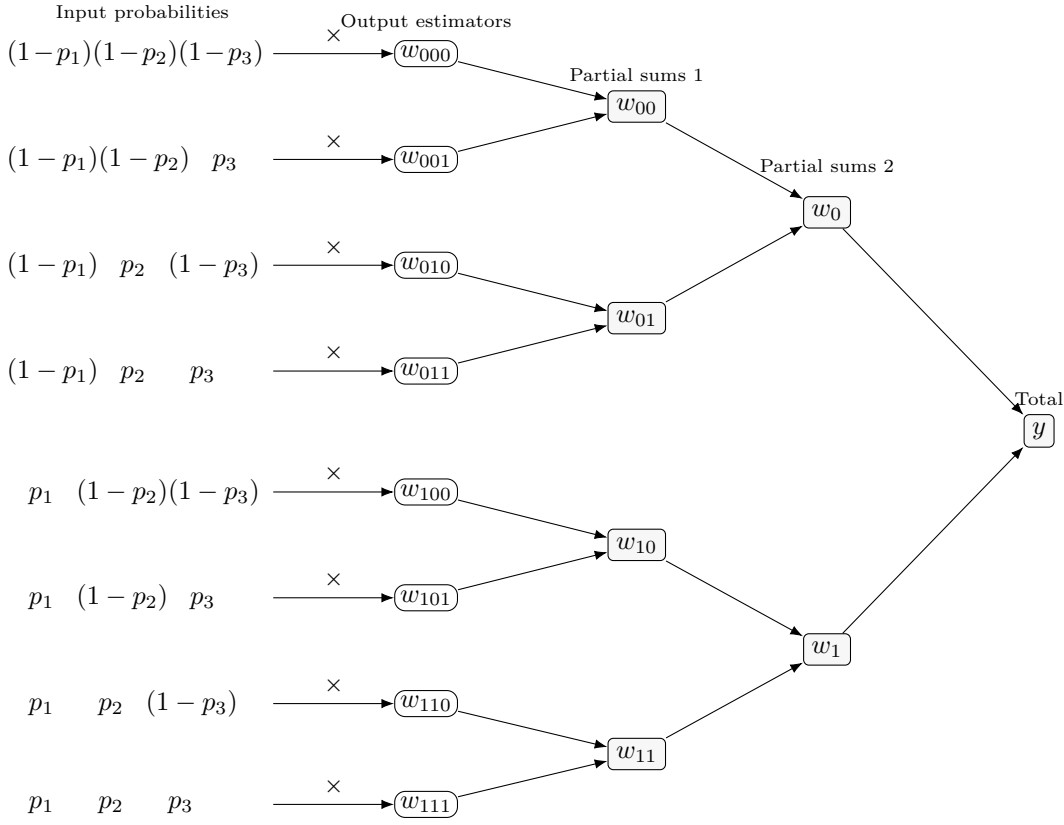

Figure 13: Computation graph of the forward kernel of a differentiable logic gate with arity $n = 3$. The tree-like accumulation order gathers terms with similar probability weights and thus mitigates numerical instability.

### B.6.1 COMPUTATIONAL FEASIBILITY

Even for double-precision floating point computations and fully unrolled single scalar assignment instructions, the register usage of the forward and backward kernels for gate arity 6 remains below the maximum 255 registers per thread on modern NVIDIA GPUs. By contrast, OP DLGNs exceed this boundary for gate arity 3 already, where the number of parameters alone becomes 256.

Still, the number of registers is evidently larger than for arity 2 kernels, and we do need to decrease the number of threads per thread block by a factor of up to 4 for the backward kernels to meet the register constraints of an NVIDIA streaming multiprocessor. Moreover, the computation graph of each logic gate deepens by a factor of 3. The overall runtime per training step thus increased by a factor of $\tilde{8}$.

### B.6.2 ANALYTICAL AND NUMERICAL GRADIENT STABILITY

Computing the forward and backward pass boils down to a computing a weighted sum of the output estimators $\omega$. Recall Equation (11) for the case $n = 2$. At low-level implementation in the forward and backward kernels, this accumulation is performed in a tree-like accumulation order which we visualize in Figure 13 for $n = 3$. This tree structure is not only a natural choice to minimize instruction dependencies and latency. It also ensures that terms have a similar magnitude when added together. For example, no matter how extreme $p_1$ and $p_2$ concentrate towards 0 and 1, the tree accumulation consecutively accumulates "nearest-neighbour" weights whose probability term only differs in $p_3, (1 - p_3)$. That way, the risk of precision loss due to numerical instability increases only moderately with higher arity.

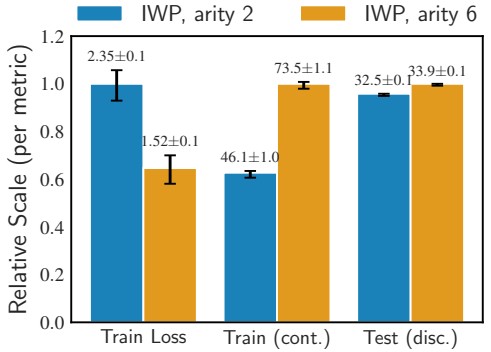

(a) Optimization metrics after 250,000 training steps.

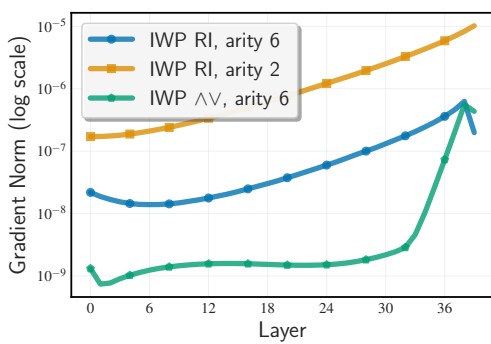

(b) Initial gradient norms for the first batch.

Figure 14: Juxtaposition of the gradient norm and prediction accuracies of IWP DLGNs with arity $n = 2$ and $n = 6$.

Consequently, the average gradient norm remains similarly stable as for arity $n = 2$, both for the residual initializations and the AND-OR initialization (cf. Figure 14b).

### B.6.3 OPTIMIZATION EFFICIENCY

Increasing the gate arity from 2 to 6 drastically raises the model's ability to fit the training dataset in a small number of steps. However, the test accuracy improves only slightly (cf. Figure 14a). We emphasize that this is not due to a high discretization error but to a high generalization error.

However, increasing the arity to 6 unleashes another 8.4 convergence speedup. Figure 15 showcases that the arity 2 IWP DLGN reaches its best test accuracy after 210000 steps, while the arity 6 IWP DLGN already surpasses this accuracy after 25000 training steps only. However, this benefit in convergence speed remains on par with the eightfold longer runtime of each training step that was reported in Section B.6.1. However, the current CUDA kernels are just naive implementations without common optimizations such as pipelining or improved memory access patterns. Such optimizations could further improve runtime performance and eventually render arity 6 DLGNs more efficient.

Another caveat is that the arity 6 model is larger, because each neuron in the arity 2 model only corresponds to a lookup-table (LUT) with 2 inputs, of which multiple can be merged into a single LUT-6 during hardware embedding.

### B.7 REMAINING ARCHITECTURAL WEAKNESSES OF DLGNS

#### B.7.1 RANDOMIZED CONNECTION TOPOLOGY FAILS TO EXPLOIT STRUCTURE IN ENCODING

To convert the real-valued inputs $x \in [0, 1]$ to binary encodings, Petersen et al. (2022) adopt the thermometer encoding $x_{th} := (x > t_1, t > t_2, \ldots, x > t_k)$, where $t_i = i/k + 1$ are evenly spaced thresholds in $[0, 1]$ (Carneiro et al., 2015). The number of thresholds directly determines the discretization resolution, and an increase should hence further decrease the approximation error. For a standard CNN architecture (cf. Figure 17), there is indeed a noticeable improvement. But for the convolutional DLGN, such an improvement fails to appear (cf. Figure 16).

Since the DLGN architecture does not lag behind the CNN architecture in expressivity (cf. Figure 16a), we hence locate the bottleneck in the random, fixed initialization of connections. In the early layers, an encoding-aware connection heuristic or even learned connections as in Bacellar et al. (2024) might overcome this limitation.

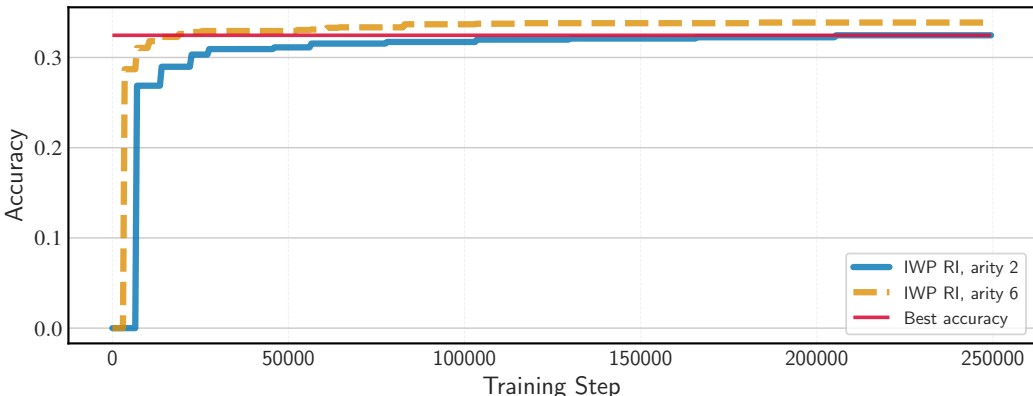

Figure 15: Increasing the gate arity from 2 to 6 yields another 8.4x convergence speedup. We still consider the CIFAR DLGN with 3-fold depth. We juxtapose the best discretized accuracy that has been achieved so far during training for both arities. The best test accuracy of the arity 2 DLGN is indicated by the red roofline.

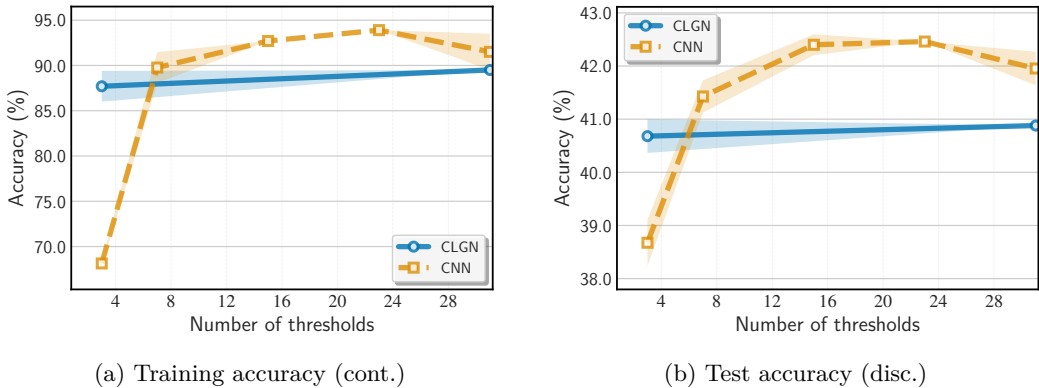

(a) Training accuracy (cont.)          (b) Test accuracy (disc.)

Figure 16: Approximation improvement for increased resolution in the thermometer encoding for the CDLGN and a standard CNN architecture.

### B.7.2    EXTRACTING CHROMATIC INFORMATION FROM COLOR SPACE ENCODING

The randomized connection topology not only hinders from exploiting the structure in the binary encoding of each real-valued channel, but also to mix information from different channels. This might explain why switching from the standard RGB color space encoding used in Petersen et al. (2022; 2024) to a HSV color space improves performance on ImageNet32 (cf. Table 3). While the hue channel in HSV solely encodes the chromatic information, all channels in RGB need to be combined to obtain a proper estimate of the chromatic value.

## C    IMPLEMENTATION DETAILS FOR REPARAMETRIZATION

### C.1    ESTIMATION FUNCTION OF LOGIC GATE OUTPUTS

While the OP slightly benefited from weight decay (Petersen et al., 2024), we have to disable it for our IWP. The reason is that for both the sigmoid and sinusoidal estimators, a weight $w$ close to 0 corresponds to an undecisive gate output $\omega \simeq 0.5$. Weight decay hence actively encourages a high discretization error and entails weaker performance at inference.

| Layer # | Description |
|---------|-------------|
| 1 | Conv2D (in_channels=..., out_channels=256, kernel=3x3, padding=1) |
| 2 | ReLU |
| 3 | Conv2D (in_channels=256, out_channels=512, kernel=3x3, padding=1) |
| 4 | ReLU |
| 5 | MaxPool2D (kernel=2x2, stride=2) |
| 6 | AdaptiveAvgPool2D (output_size=1x1) |
| 7 | Flatten |
| 8 | Linear (in_features=512, out_features=256) |
| 9 | ReLU |
| 10 | Linear (in_features=256, out_features=100) |

Figure 17: CNN Architecture built via `torch.nn` for CIFAR-100 classification in Figure 16.

| Color space | Test accuracy (cont.) | Test accuracy (disc.) |
|-------------|----------------------|----------------------|
| RGB | $5.25 \pm 0.05$ | $5.11 \pm 0.03$ |
| HSV | $6.25 \pm 0.05$ | $6.15 \pm 0.06$ |

Table 3: Test accuracies (continuous and discretized) for CIFAR DLGNs with 3-fold depths on ImageNet32 for different color space encodings. Mean and standard deviation are computed from two seeds. We juxtapose the default RGB to the HSV color space, which isolates chromatic information in a single channel.

### C.1.1 SINUSOIDAL ESTIMATOR

Heavy-tail initializations as motivated in Section 3.3 can be adjusted by adjusting shift and variance of the normal initialization. We choose $\mu = 1.2$ and $\sigma = 0.25$, which results in a distribution like Figure 18b.

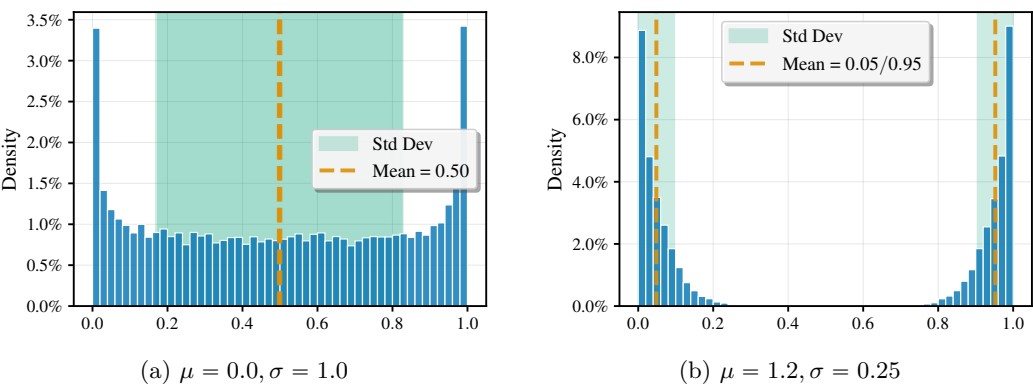

(a) $\mu = 0.0, \sigma = 1.0$  (b) $\mu = 1.2, \sigma = 0.25$

Figure 18: Initial distribution of coefficients $\omega_{ij}$ when initialized with and without a RI for the sinusoidal output estimator, i.e. $\omega_{ij} = 0.5 + 0.5 \cdot \sin(\Omega_{ij}), \Omega_{ij} \sim \mathcal{N}(\mu, \sigma)$.

### C.1.2 SIGMOID ESTIMATOR

For the sigmoid estimator that is more commonly used in logistic regression, we can similarly adopt heavy-tail initializations by shifting the weights $\Omega_{ij}$ by 3.0 (cf. Figure 19).

### C.1.3 PERFORMANCE AND GRADIENT STABILITY

Although the sigmoid function has been widely adopted for its theoretically desirable properties, its gradients vanish faster for large input values. At the same time, the periodicity of the sinusoidal estimator avoids such a dead end. But for a heavy-tail initialization as in

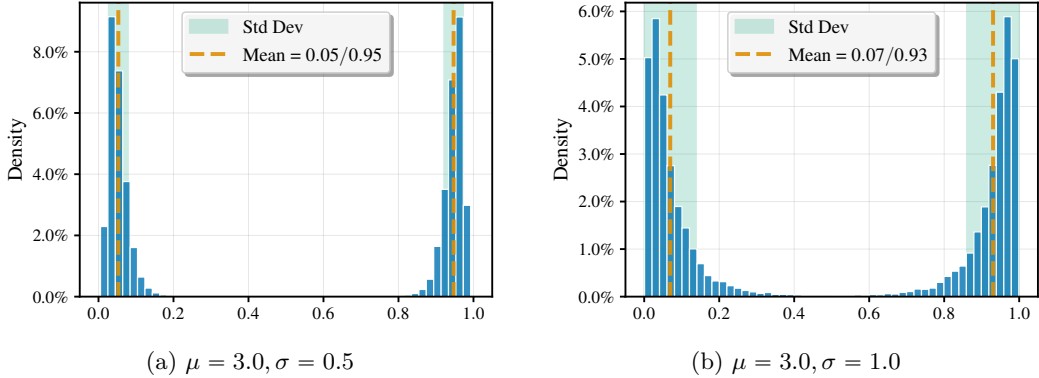

(a) $\mu = 3.0, \sigma = 0.5$           (b) $\mu = 3.0, \sigma = 1.0$

Figure 19: Initial distribution of coefficients $\omega_{ij}$ when initialized with and without a RI for the sigmoid output estimator, i.e. $\omega_{ij} = (1 + \exp(\Omega_{ij}))^{-1}, \Omega_{ij} \sim \mathcal{N}(\mu, \sigma)$.

Figure 19a that does not shift the weights too strongly into this flat region, the gradient is still sufficiently high to allow deviation from the initialization region. Although the gradient norm is initially smaller across layers compared to the sinusoidal (cf. Figure 20b), we observe that the gradient norm recovers quickly after a few batches only and approaches the curve of the sinusoidal. However, logic gate networks with a sinusoidal estimator still achieve slightly superior accuracies (cf. Figure 20a), which is why we eventually stuck with them.

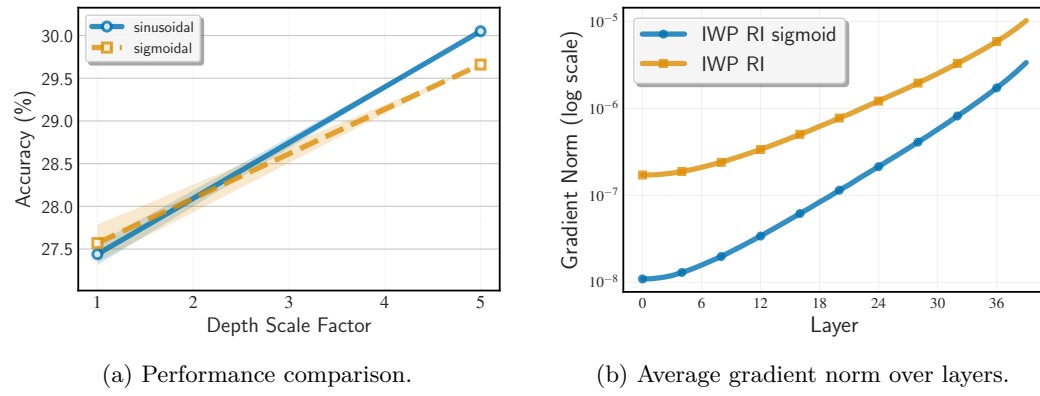

(a) Performance comparison.          (b) Average gradient norm over layers.

Figure 20: Performance and gradient stability comparison for the sigmoid and the sinusoidal gate output estimator.

## D  EXPERIMENTAL SETUP

### D.1  SCALING DLGNS IN DEPTH

As the original DLGN uses a uniform width for all logic layers, we can simply scale the DLGN in depth by placing $D$ logic layers everywhere a logic layer was placed in the original architecture.

For the CDLGN architecture, we place a block of $D$ convolutional logic layers instead of one, but apply the max pooling layer only once at the end. Because the kernel size, padding, and stride in the original architecture (Petersen et al., 2024, Sec. 3.4) preserve the spatial dimensions of the data tensors, no further adjustments are needed. As for the original DLGN, channel increases and decreases are only performed once at the initial and final convolutional logic layer of the block. Finally, we do not restrict the CDLGN architecture to

partition the range of channels into separate, independent streams as motivated by Petersen et al. (2024, Sec. 3.4) for more efficient hardware embeddings and data movement during training, but allow connections to be formed between any combination of channels.

## D.2 DEVIATIONS FROM ORIGINAL EXPERIMENTAL SETUP

Scaling DLGNs in depth increases the overall computational cost for training. To ensure that gradient descent converges even for deep models, we increase the number of training iterations from 200,000 to 250,000. Furthermore, when training sufficiently deep CDLGNs, GPU memory limitations hinder us from loading batches of original size 100. To ensure comparable optimization conditions for these models, we hence employ batch accumulation for depths $D \geqslant 4$. In particular, we accumulate four batches of size 25 in one backward pass for depths $D = 4, 5$, and tested that it behaves identically to training on the original batch size 100.

### D.2.1 10-FOLD CLASS INCREASE FOR CIFAR-100

The 10-fold class increase can be encountered in two different ways: On the one hand, one could keep the final logic layer unchanged and accumulate 10 times fewer gate outputs per class in the GroupSum layer. Petersen et al. (2024) proposed the heuristic to shrink the softmax temperature by the square root of the class increase $\sqrt{10}$ in such a case for optimal performance. On the other hand, one could increase the final logic layer to 10-fold width, which does not change the number of gate outputs per class and hence does not require any temperature adjustment.

For both the DLGN and CDLGN, increasing the width 10-fold further improves performance (cf. Figure 21). At the same time, decreasing the temperature as proposed by Petersen et al. (2024) indeed maintained optimal performance, with only minor changes when decreasing the temperature further by $\sqrt{10}$ (cf. Figure 21b).

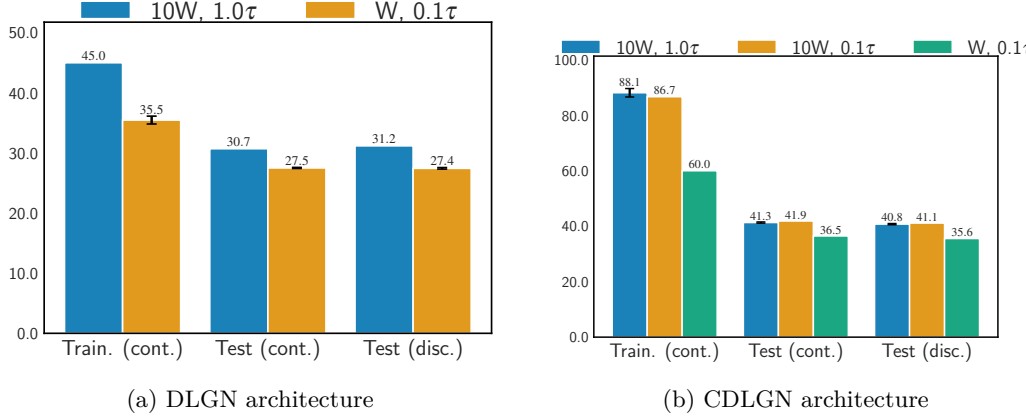

(a) DLGN architecture      (b) CDLGN architecture

Figure 21: Performance comparison for different final logic layer widths and temperatures in class score accumulation.

But for our experiments, we do not consider the choice between keeping the width and decreasing the temperature or increasing the width to keep the absolute temperature a crucial one. The reason is that we merely focus on different parametrizations of each neuron that leave their functional characteristics unchanged. We hence do not expect the trends that we observe when scaling these networks in depth to alter across these slightly varying widths of the final logic layer. To cover both options, we choose to keep the width and decrease the temperature for the DLGN, and keep the temperature and increase the width for the CDLGN.

### D.3 Runtime Measurements

Our objective is to assess the runtime performance of both parametrizations in a comparable way. To rule out possible discrepancies that are unrelated to the IWP, we build a Python subclass of the original classes for logic layers that can execute both our IWP and the OP. At runtime, a Boolean variable determines which parametrization is chosen. Apart from the different weight initialization and the invocation of the custom autograd function, the footprint of this algorithm on the machine is hence identical. We measure the past nanoseconds for an entire forward and backward pass each, and enforce synchronization at both time points to ensure that the total computation of all streams on the GPU is captured in the time measurements.

To quantify uncertainty, we take measurements for 20 different, randomly sampled batches.

### D.4 DLGN architectures for other datasets

#### D.4.1 ImageNet32

For ImageNet, we use the same DLGN architecture and the same data augmentations as for CIFAR-100. To accomodate the 10-fold increase in classes, we downscale the temperature by 10 as proposed in Petersen et al. (2022). For both the CNN and DLGNs, the RGB channel values are quantized with a 7-threshold thermometer encoding. Note that we only used 3 thresholds for CIFAR-100. However, Figure 16 demonstrates that increasing the number of thresholds benefits the CNN more than the DLGN architecture.

#### D.4.2 CIFAR-10

For CIFAR-10, we use the baseline CIFAR-10 M LGN architecture from Petersen et al. (2022) with 4 layers of $128,000$ neurons each, just as we have done for CIFAR-100.

#### D.4.3 CIFAR-100

For CIFAR-100, we presented the results for the DLGNs with 3-fold depth.

#### D.4.4 MNIST and Fashion-MNIST

For the simpler gray-scale datasets, we use 1-threshold thermometer encoding and a smaller width of $32,000$ neurons, as was done in Petersen et al. (2022). The resulting model is shown in Figure 22.

| Layer # | Description |
|---------|-------------|
| 1 | Encoding (type=thermometer, resolution=1) |
| 2 | LogicLayerIWP (in_features=784, out_features=32000) |
| 3 | LogicLayerIWP (in_features=32000, out_features=32000) |
| 4 | LogicLayerIWP (in_features=32000, out_features=32000) |
| 5 | LogicLayerIWP (in_features=32000, out_features=32000) |
| 6 | GroupSum (k=10, $\tau = 100.0$) |

Figure 22: DLGN architecture for MNIST and Fashion MNIST.

#### D.4.5 WMT'14

For the WMT'14 DLGNs, we use the architectures used by Bührer et al. (2025). The only changes we make are: 1) We make the model use our IWP rather than the original OP. 2) and 3) We vary the vocabulary and context sizes. See complete results in Section E.

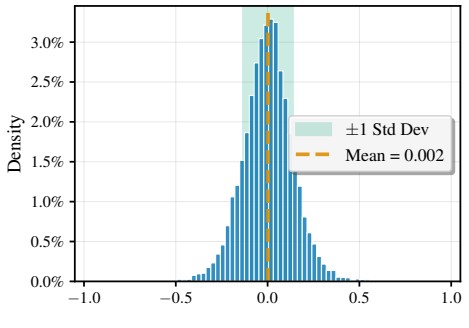
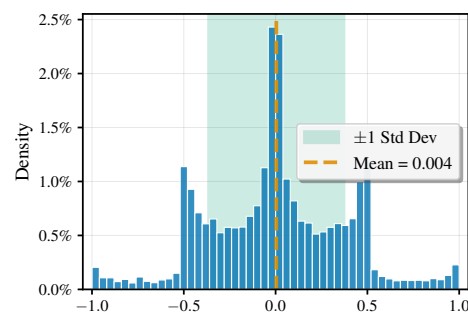

(a) Logit initialization variance $\sigma^2 = 1.0$       (b) Logit initialization variance $\sigma^2 = 16.0$

Figure 23: Self-cancellations in the sign-symmetric sum $\sum_{i=1}^{8}(\omega_i - \omega_{\neg i})$ concentrates the gradients around zero (cf. 23a), as long as the initialization variance $\sigma$ of the logits is not overly high (cf. 23b). Empirical distribution for $N = 10^4$ gradient samples $\frac{\partial g(p,q)}{\partial p}$ with $q = 0.5$.

## E    NLP TASK: ENGLISH TO GERMAN WMT'14

We show in Table 4 the corpus BLEU scores for discretized and relaxed DLGNs under OP and IWP for the English to German translation task from WMT'14 (Bojar et al., 2014a). See the experimental setup in Section D.4. We see that for almost all the parameters, IWP significantly outperforms OP. It is only for vocabulary and context sizes of 16,000 and 32, respectively, that the difference is not significant. Notably, we see that IWP decreases less when going from the relaxed to the discretized setup.

| | | | BLEU | |
| | | | Discretized | Relaxed |
| Vocabulary size | Context size | Parameterization | | |
|---|---|---|---|---|
| 8,000 | 16 | OP | $15.11 \pm 0.64$ | $17.91 \pm 0.27$ |
| | | IWP | $17.38 \pm 0.04$ | $19.09 \pm 0.53$ |
| | 32 | OP | $13.61 \pm 3.21$ | $16.71 \pm 0.54$ |
| | | IWP | $17.36 \pm 0.52$ | $17.77 \pm 0.06$ |
| 16,000 | 16 | OP | $10.65 \pm 0.20$ | $15.17 \pm 0.12$ |
| | | IWP | $14.40 \pm 0.60$ | $15.63 \pm 0.11$ |
| | 32 | OP | $10.22 \pm 0.28$ | $14.61 \pm 0.82$ |
| | | IWP | $13.50 \pm 0.41$ | $14.91 \pm 0.21$ |

Table 4: Performance (corpus BLEU ↑) of DLGNs with OP or IWP on the WMT'14 English to German translation task. We use the LGNs from Bührer et al. (2025). We measure the BLEU score using the `sacrebleu` Python library. We measure the metrics across two different seeds and report the mean and standard deviation.

## F    THEORETICAL ANALYSIS OF PARAMETRIZATION

### F.1    VANISHING GRADIENTS IN OP

Although RIs successfully suppress vanishing gradients, the symmetric parametrization still traps them in a dichotomy between gradient stability and stalling optimization towards other gate functions. On the one hand, Petersen et al. (2024)'s choice of $z = 5$ will sufficiently preserve the gradient norm, as it will decrease by at most $\frac{e^z - 1}{e^z + 15} \approx 0.9$. On the other hand, already slightly decreasing to $z = 3$ would again elicit vanishing gradients after only a few layers, as $\frac{e^z - 1}{e^z + 15} < 0.55$.

## F.2 ALGEBRAIC INTERPRETATION OF THE IWP

To understand the redundancies from an algebraic viewpoint, we can regard the space of binary functions $\mathcal{G}_2 := \{G : \{0,1\}^2 \to \{0,1\}\}$ as a vector space over the field $\mathbb{Z}_2$. Firstly, seven of the eight aforementioned negation symmetries correspond to linear dependencies $0 = G_i + G_{\neg i} + 1$ between elements in $\mathcal{G}_2$. Secondly, the redundancy that led to the suboptimal rounding in the example on the discretization error can be captured in the linear dependency $0 = G_3 + G_6 + G_8 + 1$.

## F.3 MINIMAL ROUNDING ERROR OF THE IWP

When rounding the gate estimator $g_\omega$ to a logic gate $g_\alpha$, we round each output estimator $\omega_{ij}$ to its closest binary number $\alpha_{ij} := \arg\min_{b \in \{0,1\}} |\omega_{ij} - b|$.

This achieves a minimal discretization error $\|g_\omega - g_\alpha$ in terms of any Minkowski norm $\|f - g\|_p := \sqrt[1/p]{\sum_{x \in \{0,1\}^2} |f(x) - g(x)|^p}$, because for any binary input $x = (i,j)$, the term $|g_\omega(x) - g_\alpha(x)| = |\omega_{ij} - \alpha_{ij}| = \min_{b \in \{0,1\}} |\omega_{ij} - b|$ by definition.

## F.4 REMAINING CAUSES OF VANISHING GRADIENTS IN IWP

Even with heavy-tail initializations that concentrate the $\omega_{ij}$ close to $0, 1$, destructive interference between gradient signals can still arise for precisely three reasons. Still, all of them are out of the control of the parametrization.

The first reason is destructive interferences that arise from the probabilistic relaxation of the Boolean functions. For example, for binary inputs $(1,1)$, the gradient of the OR function $g_8(p,q) = p + q - pq$ will be 0 for both inputs. We obtain a symmetric case with input $(0,0)$ and the AND function $g_2(p,q) = pq$.

Opposed to that, the remaining two reasons both relate to the parameter initialization of the DLGN architecture. We divide them into cancellations inside a neuron and between neurons.

Inside a neuron, cancellations can arise if the two terms in 12 have different signs. This happens precisely if $\omega_{11} > \omega_{10}$ and $\omega_{01} < \omega_{00}$, or vice versa, which holds only if the relaxation is close to the XOR function $g_7(p,q) = p + q - 2pq$ or its negated counterpart NXOR. Similar to the first reason, this behaviour is not problematic and even intended as long as the inputs carry information about the desired output. If the gate outputs $\omega_{ij}$ are close to $0, 1$, the gradient norm will remain close to 1. But in the case of low information, where both inputs $p, q \simeq 0.5$ are highly uncertain, the gradients of the probabilistic surrogate of XOR and NXOR will both collapse to 0 and annihilate the gradient signal. Depending on the logic gate distribution, this undesirable scenario will, however, inevitably occur as we scale logic gate networks in depth (cf. Figure 27). A heavy-tailed initialization of the logic gate distribution alone does not suffice to prevent this. In particular, we will observe later that even RIs suffer from this information collapse. But in theory, this is only problematic if XOR functions are present in the network, which is not the case for RIs.

Finally, even if we can avoid cancellations inside a neuron, gradients from different neurons might still cancel when they pass the same neuron. Because of the negation symmetry in Boolean functions, a parameter initialization that treats each function and its negated counterpart independently will result in sign-symmetric gradients across different neurons during backpropagation. If the gate output of a neuron is used as the input of multiple subsequent neurons, this gate will receive a sum of sign-symmetric partial derivatives. The more gates this neuron is connected to, the more this sum will concentrate at 0.

### F.4.1 HEAVY-TAIL, NEGATION-ASYMMETRIC INITIALIZATIONS

We maintain that an ideal initialization scheme should satisfy three properties to scale logic gate networks in depth: heavy tail, information preservation, and negation asymmetry.

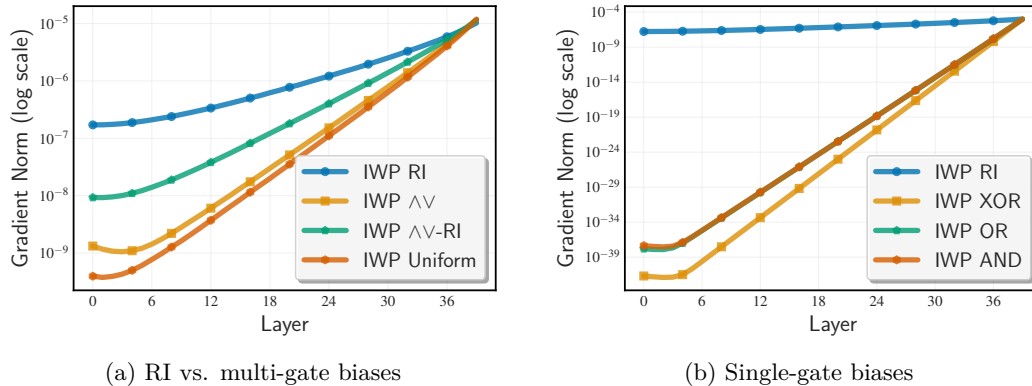

(a) RI vs. multi-gate biases

(b) Single-gate biases

Figure 24: Gradient norm decrease for different heavy-tail initializations of an IWP DLGN with 40 layers. While RIs stand out as the only stable single-gate bias, other multi-gate biases also retain stable gradients.

The normal initialization $\Omega_{ij} \overset{i.i.d.}{\sim} \mathcal{N}(0,1)$ violates all of these properties. The resulting coefficients $\omega_{ij}$ will concentrate symmetrically around 0.5 and evoke vanishing gradients, as Figure 24a illustrates.

First of all, one could ensure a heavy-tail distribution of the coefficients $\omega_{ij}$ at 0 and 1 by shifting the normal distribution in a negative or positive direction. The overall sign combination in $\Omega_{ij} \overset{i.i.d.}{\sim} \mathcal{N}(\pm\mu_{ij}, 1)$ hence attributes a high initial bias towards one of the sixteen logic gate functions. Choosing the pass-through gate $G_4(A, B) = A$ for all neurons recovers the idea of RIs.

Indeed, if we restrict ourselves to choosing only a single function for all neurons, RIs are the only viable approach. While the constant functions have no gradient anyway, the AND, OR, and XOR functions alone rapidly concentrate the intermediate feature distribution to $1, 0$, and $0.5$, as Figure 27 exemplifies. At that point, their gradients collapse to 0 and stifle any information in the input. In terms of our three necessary properties, these initializations fall short of information preservation.

On the other hand, the pass-through gate $G_4$ does not change the input value $p$, and maintains a gradient of 1 with respect to that input $p$, independent of what value $p$ takes. However, as we increase the model depth, the intermediate feature distribution will also collapse to 0.5 with RIs (cf. Figure 3). This is because even small initial uncertainties in the coefficients, i.e. $|\alpha_{ij} - \omega_{ij}| \simeq 0.05$, will accumulate over the layers. But because no gate is initially close to the XOR functions when employing RIs, this high uncertainty in later layers is harmless. On the contrary, we will discuss in the following subsection F.4.2 why this increasing uncertainty can even benefit the optimization of deep logic gate circuits.

For heavy-tail initializations that bias towards a single function in all neurons, RIs are hence indeed the unique scalable choice. But we might also combine multiple logic gate functions into a heavy-tail initialization. In the extreme case, each logic gate could bias towards one of all sixteen functions with uniform probability $1/16$. But this brings us back to the third and last property, namely, negation asymmetry.

Allowing both a Boolean function and its negated counterpart will provoke cancellations if sign-symmetric partial derivatives merge during backpropagation. Fortunately, this condition only holds for architectures with drastically increasing width between layers. For the architecture of Petersen et al. (2022) with uniform width, even negation-symmetric initializations such as the uniform initialization will retain sufficiently stable gradients (cf. Figure 24a).

But this might not hold in general. Formally speaking, any subset of the binary functions $G \subseteq \mathcal{G}_2$ that does not contain a function and its negated counterpart is a feasible negation-asymmetric subset. In particular, such a subset can be obtained by fixing one output to 0 or

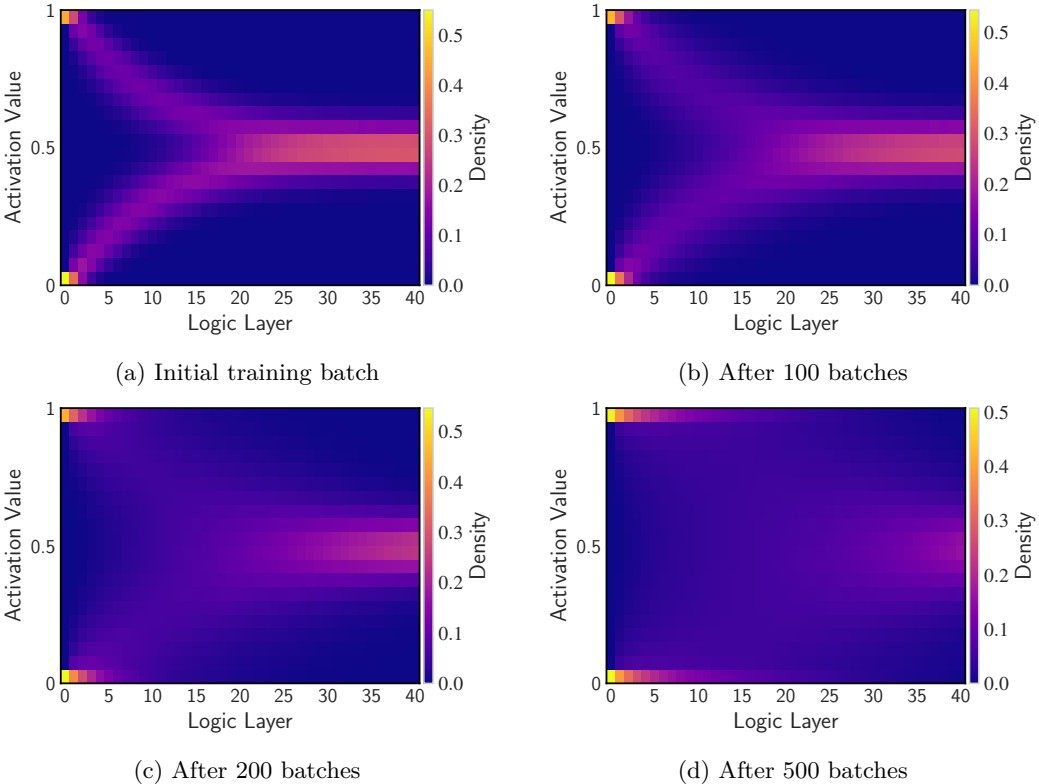

(a) Initial training batch      (b) After 100 batches

(c) After 200 batches      (d) After 500 batches

Figure 25: Distribution of intermediate gate outputs of an IWP DLGN with RIs. Measurements were taken at different timesteps over the course of training, where each batch comprises 100 CIFAR-100 images.

1 and taking half of the binary functions that coincide with this mapping. For example, by enforcing $00 \mapsto 0$, we admit the constant 0, the two pass-through gates, three AND functions, one OR function, and the XOR function. Therefore, an alternative to the RI is to combine the AND and OR functions into an AND-OR initialization. Indeed, the complementary concentration behaviour of the AND and OR functions avoids the information collapse at 0.5 that RIs inevitably entail. Instead, Figure 27e depicts how the feature distribution balances at values close to 0 and 1, and hence reduces the uncertainty in the signal in later layers. However, this alone does not render the AND-OR initialization more desirable than RIs. Conversely, while a collapse at 0.5 might be harmful in general, we explain in the next section why it actually benefits the optimization process in the case of RIs.

### F.4.2 RESIDUAL INITIALIZATIONS DELAY FEATURE LEARNING AT LATER LAYERS

When initializing all neurons with a pass-through gate, Figure 3 displays how the features eventually concentrate at 0.5 at later layers. At those layers, it holds that $1 - p \simeq p \simeq q \simeq 1 - q$, hence the gradient update $\frac{\partial \mathcal{L}}{\partial \omega_{ij}} = \frac{\partial \mathcal{L}}{\partial g_\omega} \frac{\partial g_\omega}{\partial \omega_{ij}}$ is roughly equal for all $i, j$. Because of that, the neurons in the later layers will maintain their pass-through function until the uncertainty reduces sufficiently. This pass-through enforcement at the later layers allows the network to begin with optimizing the earlier layers first. The more the earlier neurons approach specific gates, the more declines the uncertainty of their outputs, allowing the later layers to refine their functionality. Practically, the model first optimizes a shallow logic gate circuit and increasingly advances this circuit in depth over time. Figure 25 showcases this consecutive gate collapse at earlier layers and uncertainty decrease at later layers over the course of training. This implicit organization of feature learning not only tames the overall discretization error but will also lead to faster convergence. On the contrary, the initial feature distribution of the AND-OR initialization will allow neurons at all layers to update

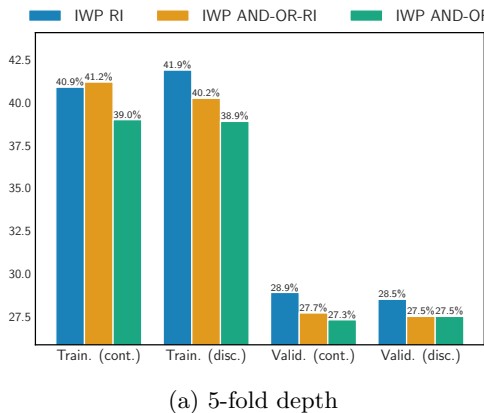
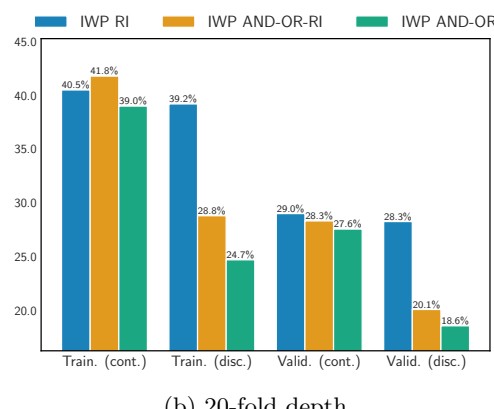

| (a) 5-fold depth | (b) 20-fold depth. |

Figure 26: In contrast to the structured layer optimization of DLGNs with RIs that steadily maintains a low discretization error, the simultaneous layer optimization for AND-OR initialization drastically increases the discretization error and harms overall performance.

their coefficients in a non-uniform fashion at the same time. The drawbacks of such a more chaotic optimization process become noticeable as we scale those networks in depth.

While the discretized accuracy of both initializations remains similar for shallower logic gate networks with 4 or 20 layers, scaling these networks to 80 layers exposes a clear discretization gap for the AND-OR initialization. At the same time, the RI maintains a low rounding error over the course of training and exhibits slightly superior predictive performance (cf. Figure 26). Similar drawbacks also hold for a uniform initialization or an initialization that combines AND, OR, and pass-through gates (cf. Figure 27).

To conclude, pairing our exact IWP with RIs results in logic gate networks that are scalable in depth and can harness the associated expressive benefits.

## G REGULARIZING LOGIC GATE NETWORKS

To mitigate the generalization error, we try to impose several constraints on the DLGN architecture that have benefited standard neural network architectures. Unfortunately, the methods that we have tried did not raise the test accuracies further, leaving the generalization gap an open problem. In the following, we present the measures we have taken, how we implemented them for logic gate networks, and how they impacted performance.

### G.1 DROPOUT

When applied in standard feed-forward neural networks, dropout (Srivastava et al., 2014) typically randomly zeroes neurons. For the logic gate network, the zeroing operation is, however, only a neutral operation in the algebraic sense when we apply it in the summation in the GroupSum layer. For logic gates, the zero is not a neutral element, but on equal terms with its binary complement 1. We hence decide to realise dropout by randomly masking logic gate outputs at the final logic layer. To determine which outputs are affected, we randomly select channels of the input tensor and mask the outputs of all gates that are path-connected to inputs from at least one of these channels. For all affected gates, we ensure that they receive no gradient update. Each channel or feature dimension is selected independently with a probability $p_{dropout} > 0$. This selection is repeated for every single batch in training. For $p_{dropout} = 0.02$, roughly 30,000 of the 120,000 logic gates in the final layer are masked. For $p_{dropout} = 0.05$, this number increases to 70,000, and culminates in 100,000 for $p_{dropout} = 0.1$.

However, Figure 28a shows that the test accuracies degrade with increasing dropout probability. This regularization strategy does not, hence, seem beneficial.

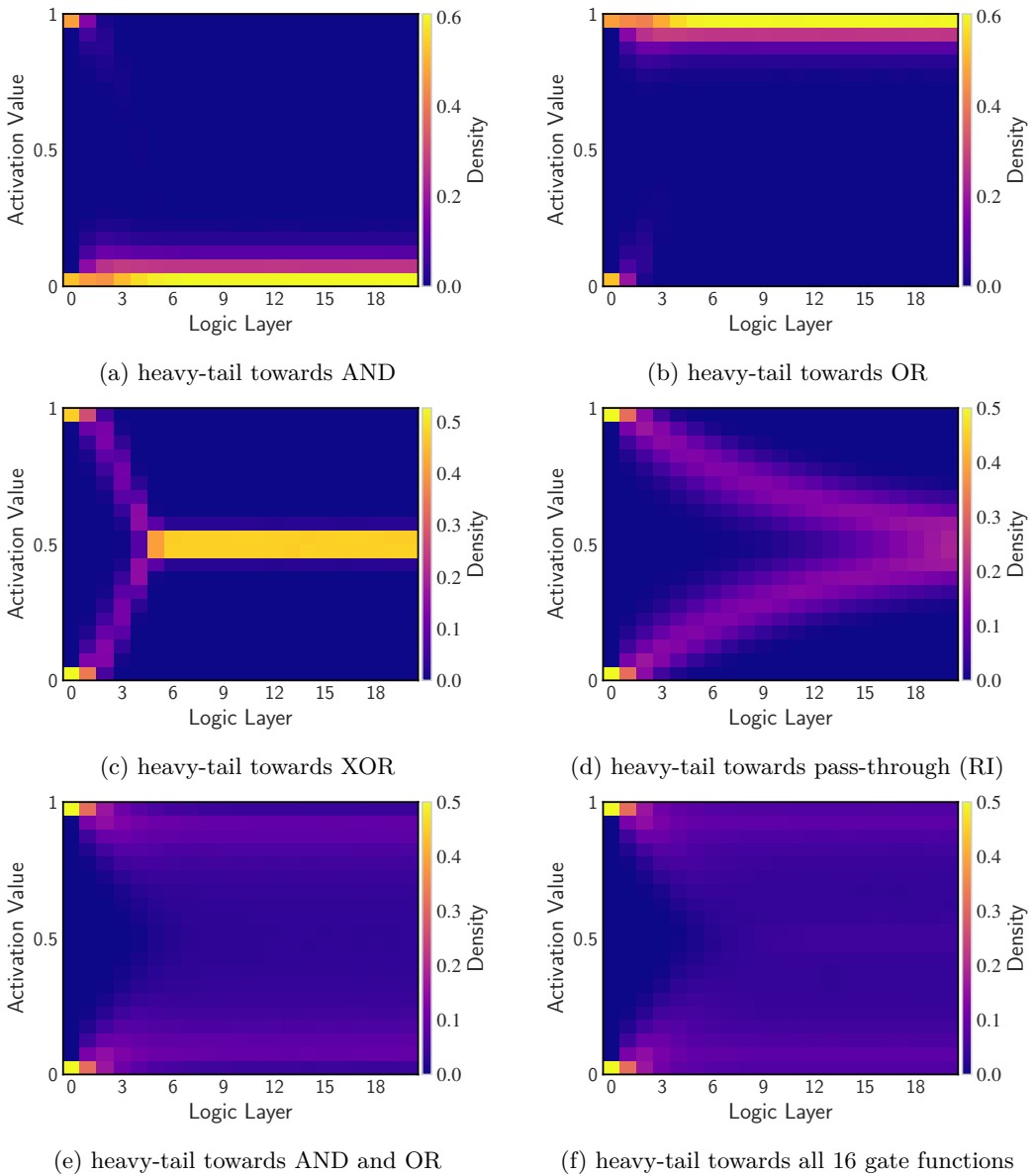

Figure 27: Initial distribution of intermediate gate outputs, averaged over 100 CIFAR-100 images, when initialized with different heavy-tail initializations.

## G.2 RANDOMIZED GATE INTERVENTIONS

Similarly, we try to randomly intervene in the output of each gates in the network with a probability $p_{intervene} > 0$. We explore several strategies to replace the actual gate output: from a simple replacement by a constant value to replacement by a random uniform $b \sim U([0,1])$ or a symmetric Bernoulli $b \sim B(0.5)$. We explore the impact of magnitude for the intervention probability $p_{intervene}$ and find $p_{intervene} = 0.05$ to yield the best results in the end. Indeed, the generalization gap narrows substantially, but the test accuracies still trail the unregularized DLGN for all intervention strategies (cf. Figure 28b).

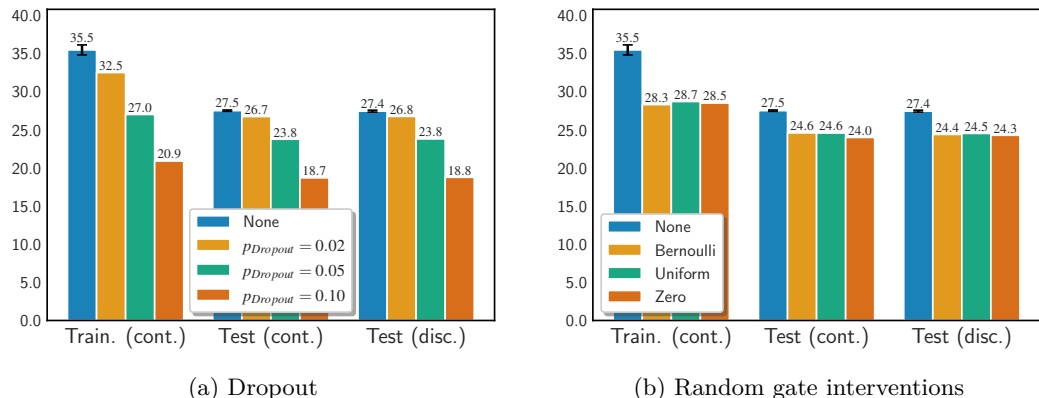

(a) Dropout

(b) Random gate interventions

Figure 28: Accuracies of the DLGN with dropout and random gate interventions.

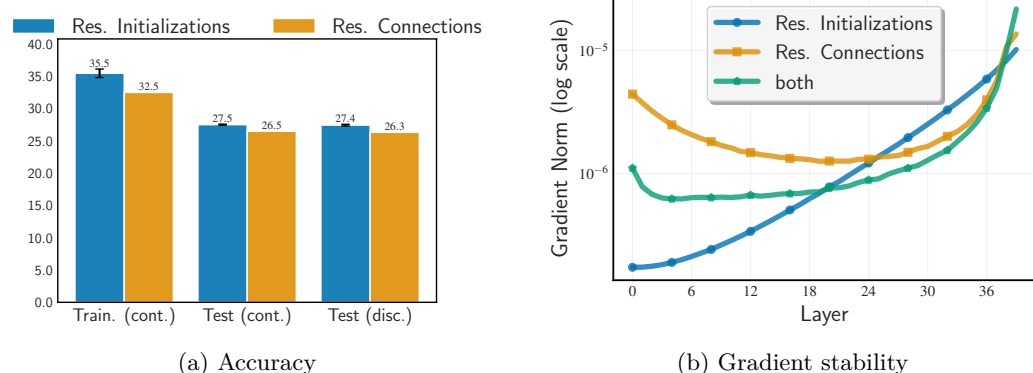

(a) Accuracy

(b) Gradient stability

Figure 29: Accuracies and gradient norms of the DLGN with residual connections compared to RIs.

### G.3 RESIDUAL CONNECTIONS

Finally, we explore if the network benefits from enforcing explicit residual connections (He et al., 2016) between layers instead of RIs. From the first to the last layer, a linearly increasing fraction of gates are fixed to directly pass their output to a unique neuron in the subsequent layer. We ensure that incoming residual streams from earlier layers are continued until the last layer. That way, each layer is guaranteed to receive a fraction of unreduced gradient signals, even when the remaining weights are not initialized with a heavy tail, but a standard Gaussian. Unsurprisingly, the gradient norms of DLGNs with residual connections are even more stable than for RIs, which still include some uncertainty in the weights $\omega_{ij}$ (cf. Figure 29b). However, Figure 29a indicates that both the training and test accuracies suffer slightly from this functional constraint. Although they half the number of learnable parameters and allow to retain gradients norms without heavy-tail initializations, residual connections do not seem to play a beneficial role for generalization.

## H  RELATED WORK

### H.1  LEARNING SINGLE LOGIC GATES

Several works have exploited that learning circuits of logic gates with more than two inputs allows to embed more functional expressivity on the same hardware (Umuroglu et al., 2020; Bacellar et al., 2024).

The reason is that a single logic gate with $n$ inputs has a VC dimension of $2^n$ (Vapnik & Chervonenkis, 1971). On the contrary, a circuit of binary logic gates with $n$ inputs has a strictly smaller discriminative power, as the VC dimension of subcircuits merely accumulates additively and not multiplicatively (Andronic & Constantinides, 2025).

On the contrary, DLGNs were practically limited to learn logic gates with very few inputs, as processing $2^{2^n}$ parameters per logic gate with $n$ inputs quickly becomes intractable. With our IWP that reduces the number of parameters to $2^n$, advancing DLGNs to process more than two inputs per gate becomes a viable option.

In contrast to our IWP, these works do not directly estimate the outputs of the logic gates. Instead, they use a different representation class and quantize this class to logic gates after training. However, these indirect representations either fall short of exploiting the expressivity of logic gates (Umuroglu et al., 2020) or are costlier to parametrize (Andronic & Constantinides, 2023; 2025). To begin with, Bacellar et al. (2024) do not relax the logic gate at all and approximate gradients via a finite difference method that accumulates all $2^n$ function values in a weighted sum. Most other works relax each logic gate to a continuous function class during training and quantize it back afterwards (Umuroglu et al., 2020; Andronic & Constantinides, 2023; 2025). Our IWP also falls within this category. However, these works differ from our IWP in that these function classes either do not completely exploit the expressivity of logic gates or require more parameters to train. On the one hand, Umuroglu et al. (2020) merely regress an affine transformation $w^T x + b$ that is fed through an activation function after batch normalization. Here, $x$ is the input vector, and $w, b$ are learnable weights and bias. Although the parameter size of each neuron grows only linearly in the number of logic gate inputs, this relaxation can also express only a small subset of Boolean functions. Andronic & Constantinides (2023) hence extends this relaxation to kernelized regression $w^T \phi(x) + b$ with a polynomial kernel $\phi$ that maps $x$ to all monomials of degree at most $D$, where $D$ is a configurable parameter. The size of $w$ hence scales to $n^D$, where $n = \dim(x)$ is the number of inputs. To completely cover the class of Boolean functions, one needed to scale $D$ to $n$ in order to incorporate the conjunction of all $n$ inputs. The resulting weights would then have dimension $n^n$, which is larger than our $2^n$. Finally, Andronic & Constantinides (2025) learn even larger neural networks within each logic gate relaxation.

## H.2 Unrelated advancements

Finally, these works contributed several advancements that do not relate to the parametrization, such as learning and simplifying the connection topology or regularization.

### H.2.1 Learning connections

Petersen et al. (2024) maintained that randomly initializing the connections between logic gate functions ab initio and leaving them fixed during training does not degrade performance. Instead, Bacellar et al. (2024) learn these connections via a softmax relaxation. This degree of freedom however comes at the cost of learnable weight matrixes whose dimensions correspond to the widths of contiguous layers.

### H.2.2 Regularization

While Andronic et al. (2025) employ pruning strategies that incorporate the connection topology of the hardware, Bacellar et al. (2024) exert regularization on the Fourier transform of each logic gate (O'Donnell, 2014).

### H.2.3 Classification head

To convert the logic gate outputs into a classification, DLGNs counts the bits for each class and outputs the class index with the highest sum. To avoid the additional overhead of embedding these operations in FPGA hardware, Bacellar et al. (2024) replace them by learnable lookup tables.

Table 5: All binary logic functions with real-valued relaxations and gradients

| id | $G_i$ | $G_i(0,0)$ | $G_i(0,1)$ | $G_i(1,0)$ | $G_i(1,1)$ | $g_i$ | $\frac{\partial g_i}{\partial A}$ | $\frac{\partial g_i}{\partial B}$ |
|----|-------|-----------|-----------|-----------|-----------|-------|-----------------------------------|-----------------------------------|
| 1 | $0$ | 0 | 0 | 0 | 0 | $0$ | $0$ | $0$ |
| 2 | $A \wedge B$ | 0 | 0 | 0 | 1 | $AB$ | $B$ | $A$ |
| 3 | $\neg(A \to B)$ | 0 | 0 | 1 | 0 | $A(1-B)$ | $1-B$ | $-A$ |
| 4 | $A$ | 0 | 0 | 1 | 1 | $A$ | $1$ | $0$ |
| 5 | $\neg(B \to A)$ | 0 | 1 | 0 | 0 | $B(1-A)$ | $-B$ | $1-A$ |
| 6 | $B$ | 0 | 1 | 0 | 1 | $B$ | $0$ | $1$ |
| 7 | $A \oplus B$ | 0 | 1 | 1 | 0 | $A + B - 2AB$ | $1-2B$ | $1-2A$ |
| 8 | $A \vee B$ | 0 | 1 | 1 | 1 | $A + B - AB$ | $1-B$ | $1-A$ |
| 9 | $\neg(A \vee B)$ | 1 | 0 | 0 | 0 | $1 - A - B + AB$ | $-1+B$ | $-1+A$ |
| 10 | $\neg(A \oplus B)$ | 1 | 0 | 0 | 1 | $1 - A - B + 2AB$ | $-1+2B$ | $-1+2A$ |
| 11 | $\neg B$ | 1 | 0 | 1 | 0 | $1 - B$ | $0$ | $-1$ |
| 12 | $B \to A$ | 1 | 0 | 1 | 1 | $1 - B + AB$ | $B$ | $-1+A$ |
| 13 | $\neg A$ | 1 | 1 | 0 | 0 | $1 - A$ | $-1$ | $0$ |
| 14 | $A \to B$ | 1 | 1 | 0 | 1 | $1 - A + AB$ | $-1+B$ | $A$ |
| 15 | $\neg(A \wedge B)$ | 1 | 1 | 1 | 0 | $1 - AB$ | $-B$ | $-A$ |
| 16 | $1$ | 1 | 1 | 1 | 1 | $1$ | $0$ | $0$ |

