# OpenReview forum: "Light Differentiable Logic Gate Networks"
_ICLR.cc/2026/Conference — ICLR 2026 Poster_

### Official Review · Reviewer_Pghb · 2025-10-27

**Soundness:** 2
**Presentation:** 3
**Contribution:** 2
**Rating:** 4
**Confidence:** 4

**Summary:**

The paper introduces a new parameterization called input-wise parameterization (IWP) for Differentiable Logic Gate Networks (DLGNs), which converts the original parameterization (OP), where mixture-over-Boolean-functions parameterization is learned. IWP directly learns the numerical outcome of four input cases, i.e., (0,0), (0,1), (1,1), (1,0), leading to a 4x parameters reduction per gate. The authors claim that the new proposed parameterization can alleviate gradient vanishing and less computational costs. Experiments on CIFAR-100 show superior performance and faster computation of IWP compared to the original parameterization.

**Strengths:**

1. The analysis in discretization error, redundancy, and gradient stability of OP in Section 3 is insightful.
2. The manuscript is well-organized and easy to follow.

**Weaknesses:**

1. My major concern of the paper is the experiment. The authors only evaluate IWP on single CIFAR-100 datasets. I would expect the authors to include more diverse datasets, for example, MNIST, CIFAR-10, WMT’14, and possibly TinyImageNet. Experiments on more diverse datasets are needed to evaluate the proposed parameterization robustly.
2. My second concern is that the improvement over OP is marginal. For example, in CIFAR-100 the authors show 2% improvement with 8x faster convergence. However, there is no breakthrough in representation power, i.e., the depth scaling behaviors are unchanged compared to the OP as seen in Figure 4. However, note that the overall test accuracy is 29% while very small CNNs like resnet-20 can have 70%+ accuracy on CIFAR-100. Hence, results would be more compelling if deeper models with IWP closed the gap to standard architectures, not just DLGNs with OP.

**Questions:**

1. In line 196-197, should the argmax is the pass-through gate G4 rather than G3?
2. The expression of Equation 9 is a bit confusing to me. The equation would be more self-contained with more details like $G(k,l) = \alpha_{0,0} E_{k,l,0,0} + ... $ and $E_{k,l,i,j} = \textbf{1}_{\\{(k,l)=(i,j)\\}}$.

---

> ### Author Response · Authors · 2025-11-15
> **Experiments on more datasets in general comment above**
>
> We thank the reviewer for their thoughtful and constructive remarks. Since multiple reviewers asked for experiments on other datasets than CIFAR-100, we have posted a general comment above where we present preliminary results for ImageNet32, CIFAR-10, MNIST and Fashion-MNIST. We are conducting more experiments, but we do not want to put off the discussion until then.
> We will respond to the other remarks in detail in a separate comment as soon as possible.

---

> ### Author Response · Authors · 2025-11-20
> **Complete Answer**
>
> We thank the reviewer for raising these concerns, and approach them individually in the following.
> ### 1 - More diverse experiment results: Far more robust, consistently superior
> We have now conducted more diverse experiments, and these are our findings:
> - The IWP DLGN **consistently outperforms** the OP on various datasets, from MNIST to ImageNet32. We kindly refer the reviewer to the general comment above for details.
> - IWP DLGNs are **far more robust** than OP DLGNs against variations of hyperparameters, such as the GroupSum softmax temperature or the choice of optimizer (e.g., **12-13% better absolute accuracy** on SGD and Adadelta). For detailed experiments, please see Section 1 in our reply to Reviewer Nt7w.
>
>
> ### 2 - Representation power of DLGNs
> We emphasize that vanilla CNNs do not perform much better than DLGNs when provided with the same quantized input data.  We demonstrate this both for ImageNet32 in the general comment above and for CIFAR-100 in Figure 11 in our paper. The encoding, hence, acts as an information bottleneck that curbs the overall achievable representation power.
>
>
> Apart from that, the reparametrization of DLGNs alleviates major bottlenecks that prevented the original architecture from scaling along multiple dimensions. We want to stress two beneficial aspects of this scalability:
> - Increasing the gate arity finally becomes feasible, which yields **another >8x convergence speedup** in training steps for gate arity 6, and another accuracy improvement. For details, please see Section 4 of our reply to Reviewer Nt7w.
> - For the deeper, more recent convolutional DLGN architecture from Petersen et al. (2024), we also remark that increasing depth exposes a considerably larger performance gap than for the original architecture.
>
>
> Thanks to this robust and efficient refinement of the underlying parametrization, the remaining hurdles towards competitive performance can now be investigated at a larger scale.
>
>
> The reparametrization is surely not the only, but an important first milestone on this quest.
> For example, the randomized connection topology prevents DLGNs from leveraging structure along both the spatial and channel dimensions. Figure 11 in Appendix B.5 demonstrates the latter, while the performance increase of convolutional DLGNs in Petersen et al. (2024) confirms the former.
>
> Finally, we thank the reviewer for the attentive remarks about our manuscript and have gladly integrated the suggestions into the updated version. Please let us know if there are any further comments or inputs; we are happy to continue the discussion. If you are content with the improvements of our submission, we would appreciate it if you could reflect this in your score.

---

> > ### Comment · Reviewer_Pghb · 2025-11-24
> >
> > I appreciate the authors for providing the additional experiments on diverse datasets, they look promising when compared with DLGNs with original parameterization.
> >
> > However, one remaining concern is about the practical use of DLGNs. Even with the improved parameterization, the results (See Figure 5) show that DLGNs still do not scale well in depth or number of parameters after a certain threshold, especially compared to standard architectures like CNNs or Transformers.
> >
> > Could the authors comment on the intended practical role of DLGNs given these limitations. For example:
> >
> > - Are DLGNs mainly used for ultra-low-resource hardware where efficiency is more important than accuracy?
> > - Should we view DLGNs as an “efficiency-focused” model rather than something expected to scale like modern neural nets?
> > - Do the authors see a path for DLGNs to eventually scale competitively like other model architectures?

---

> ### Author Response · Authors · 2025-11-25
>
> ## Are DLGNs mainly used for ultra-low-resource hardware where efficiency is more important than accuracy?
>
> Yes, DLGNs are currently focused on working well in low-resource settings. However, they also allow for ultra-fast inference, so they can also work well in domains where low latency is essential.
>
> ## Should we view DLGNs as an “efficiency-focused” model rather than something expected to scale like modern neural nets?
>
> Yes and no. At the moment, DLGNs achieve lower accuracy compared to modern CNNs, but CNNs have benefited from over two decades of intensive research, engineering, and community-wide innovation. In contrast, DLGNs are a very new class of models, still in their infancy, so, unsurprisingly, there is currently a performance gap. We are not claiming that DLGNs will necessarily reach the level of CNNs. However, we expect that gap to shrink as the field of differentiable logic networks matures.
> To appreciate how early-stage models behave, it is useful to recall how CNNs performed before the major breakthroughs of the past fifteen years. A convolutional model on CIFAR-10 would typically achieve only around seventy to eighty percent test accuracy [2], far below what even modest architectures achieve today. Only through a series of foundational ideas and practical insights did CNNs become scalable and competitive.
>
> We would highlight several seminal papers/ideas that allowed models to scale:
> 1. Glorot & Bengio (2010) [5] introduced normalized initialization (later called Xavier initialization) that improved the gradient variance and gave (statistically significant) better results.
> 2. Glorot et al. (2011) [4] showed that training with ReLU yielded slightly better results; also, Krizhevsky et al. (2012) showed that ReLU allows simple convolutional models to get to 25% training error six times faster than other non-linearities on CIFAR-10 [3].
> 3. The He initialization enabled training deeper models (cf. Figure 3 in [1]). A model with 22 layers converged faster with He initialization than with Xavier initialization, and a 30-layer model could only converge with He initialization.
> 4. Residual connections allowed training much deeper models than previously. Plain 20 and 56-layer models on CIFAR-10 would get a test accuracy around 80 to 90% (cf. Figure 1 in [6]), where the deeper model performed worse (the plain model with 110 layers would get less than 40% as mentioned in the caption of Figure 6 in [6]). Meanwhile, the addition of residual connections allowed one to scale the models much further in depth, resulting in test accuracies around 93.5% for a model with 110 layers.
> 5. Dropout was found to be very beneficial at decreasing the generalization gap [11].
> 6. Batch normalization allowed training models to achieve better accuracies and to train them much faster through better convergence and a higher learning rate [8]. We would here highlight the excellent analyses by Santurkar et al. (2018) [9] and Bjorck et al. (2018) [10] of why batch normalization works.
>
> We would also like to highlight this statement from He et al. (2016) [6]: “When deeper networks are able to start converging, a degradation problem has been exposed: with the network depth increasing, accuracy gets saturated (which might be unsurprising) and then degrades rapidly. Unexpectedly, such degradation is not caused by overfitting, and adding more layers to a suitably deep model leads to higher training error.”
>
> Additionally, Bradley (2010) [7] showed in his PhD thesis that the gradient norm decreases over layer depth (cf. Figure 2.4 in [7]). These results mimic what we see in Figure 8.

---

> ### Author Response · Authors · 2025-11-25
>
> ## Do the authors see a path for DLGNs to eventually scale competitively like other model architectures?
>
> Following the answer to the previous question, yes, we believe that DLGNs will be able to scale to much better performance.
>
> For example, Yousefi et al. (2025) [12] showed that many of the gates do not converge/update from the initialization, and Petersen et al. (2024) [13] showed that residual initialization of DLGNs induces a heavy bias towards the residual gate. Thus, the models are currently only exploring a small subset of the possible logic gate networks. DLGNs also exhibit a tendency to memorize the training data (we often see 100% accuracy on the training data for CIFAR-10).
>
> Besides that, the connection topology of DLGNs is currently still fully randomized and not subject to optimization. In particular, they do not exploit the spatial structure in the feature dimensions. Figure 16 in our paper demonstrates that DLGNs still fail to exploit higher resolution inputs, contrary to CNNs. Recent advances by Fojcik et al. (2025) [14] show that optimizing the sparse connection topology can further improve the performance-to-efficiency ratio of DLGNs. We present and discuss other contemporary approaches to scale learnable logic gate networks in Appendix H of our paper. We hence expect that refining and combining these ideas can push the limits of DLGNs further.
>
> Modern neural networks benefit from (at least) two decades of research exploring how to scale them and decrease the train-test generalization gap. While many of the techniques, such as convolution, can be used to train better DLGNs, there is still a lot of work to figure out how to benefit from modern regularization methods or design new ones for DLGNs. We showed in Figure 28 that the straightforward applications of dropout to DLGNs did not lead to better results.
>
> ## References
> [1] He, Kaiming, et al. "Delving deep into rectifiers: Surpassing human-level performance on imagenet classification." Proceedings of the IEEE international conference on computer vision. 2015.
>
> [2] Krizhevsky, A. (2010). Convolutional deep belief networks on CIFAR-10. Unpublished manuscript, 1-9. URL: https://www.cs.toronto.edu/~kriz/conv-cifar10-aug2010.pdf
>
> [3] Krizhevsky, Alex, Ilya Sutskever, and Geoffrey E. Hinton. "Imagenet classification with deep convolutional neural networks." Advances in neural information processing systems 25 (2012).
>
> [4] Glorot, X., Bordes, A., & Bengio, Y. (2011). Deep Sparse Rectifier Neural Networks. Proceedings of the Fourteenth International Conference on Artificial Intelligence and Statistics, in Proceedings of Machine Learning Research. Available from https://proceedings.mlr.press/v15/glorot11a.html.
>
> [5] Glorot, X. & Bengio, Y. (2010). Understanding the difficulty of training deep feedforward neural networks. Proceedings of the Thirteenth International Conference on Artificial Intelligence and Statistics, in Proceedings of Machine Learning Research. Available from https://proceedings.mlr.press/v9/glorot10a.html.
>
> [6] He, Kaiming, et al. "Deep residual learning for image recognition." Proceedings of the IEEE conference on computer vision and pattern recognition. 2016.
>
> [7] Bradley, David M. Learning in modular systems. Carnegie Mellon University, 2010. URL: https://www.ri.cmu.edu/pub_files/2010/5/dbradley_thesis.pdf.
>
> [8] Ioffe, Sergey, and Christian Szegedy. "Batch normalization: Accelerating deep network training by reducing internal covariate shift." International conference on machine learning. pmlr, 2015.
>
> [9] Santurkar, Shibani, et al. "How does batch normalization help optimization?" Advances in neural information processing systems 31 (2018).
>
> [10] Bjorck, Nils, et al. "Understanding batch normalization." Advances in neural information processing systems 31 (2018).
>
> [11] Srivastava, Nitish, et al. "Dropout: a simple way to prevent neural networks from overfitting." The journal of machine learning research 15.1 (2014): 1929-1958.
>
> [12] Yousefi, Shakir, et al. "Mind the Gap: Removing the Discretization Gap in Differentiable Logic Gate Networks." arXiv preprint arXiv:2506.07500 (2025).
>
> [13] Petersen, F., Kuehne, H., Borgelt, C., Welzel, J., & Ermon, S. “Convolutional differentiable logic gate networks.” Advances in neural information processing systems 37 (2024).
>
> [14] Fojcik, Katarzyna, Renaldas Zioma, and Jogundas Armaitis. "LILogic Net: Compact Logic Gate Networks with Learnable Connectivity for Efficient Hardware Deployment." arXiv preprint arXiv:2511.12340 (2025).

---

> > ### Comment · Reviewer_Pghb · 2025-11-27
> >
> > Thank you for your detailed comments. I have decided to raise my rating to 6.

---

### Official Review · Reviewer_Nt7w · 2025-10-31

**Soundness:** 3
**Presentation:** 3
**Contribution:** 3
**Rating:** 6
**Confidence:** 4

**Summary:**

This paper introduces an input-wise parametrization (IWP) for differentiable logic gate networks that removes redundancy, stabilizes gradients, and reduces discretization error. It cuts parameters by 4×, speeds up backward pass up to 1.86×, and converges 8.5× faster while preserving or improving accuracy on CIFAR benchmarks.

**Strengths:**

The paper presents a highly efficient reparametrization that eliminates redundancy in logic gate networks and significantly stabilizes gradients. It is grounded in a strong theoretical analysis of vanishing gradients and discretization error. The method delivers substantial improvements in training speed and resource efficiency. It also enables deeper DLGNs to train reliably without performance collapse. Finally, the approach maintains hardware compatibility, making it practical for edge and FPGA deployment.

**Weaknesses:**

See Question Below

**Questions:**

1) Missing ablation studies on hyperparameters such as learning rate and optimizer choices. Since logic-gate networks can be highly sensitive to gradient flow and gating stability, how do different learning rates and optimizers affect the gate sharpening process and convergence behavior in IWP-based DLGNs? Additionally, how robust is the model to hyperparameter variation, and does IWP mitigate instability compared to the original parametrization (**OP**)?

2) The experiments are conducted only on CIFAR-10/100, which are datasets from the same image family and share similar structure. To test generalization of the proposed IWP method to diverse problem domains, would including simpler symbolic datasets (e.g., **MNIST**) or non-vision benchmarks provide stronger evidence that the approach generalizes beyond natural images and is not over-specialized to CIFAR distributions?

3) Results show that IWP scales effectively with depth under residual initialization (**RI**). However, the study appears limited to thermometer encoding and nearest-rounding. Could the authors kindly clarify whether the stability and discretization improvements of IWP are expected to extend to alternative binary encodings or rounding strategies, and whether similar behavior holds without residual biasing?

4) Since IWP's parameter count scales as $2^n$ for $n$-input gates, did the authors examine performance and memory implications for $n>2$? Additionally, are there any observed or expected stability issues or computational constraints as the logic arity increases, particularly in terms of gradient behavior, training efficiency, or numerical sensitivity?

---

> ### Author Response · Authors · 2025-11-15
> **Experiments on more datasets in general comment above**
>
> We thank the reviewer for their thoughtful and constructive remarks. Since multiple reviewers asked for experiments on other datasets than CIFAR-100, we have posted a general comment above where we present preliminary results for ImageNet32, CIFAR-10, MNIST and Fashion-MNIST. We are conducting more experiments, but we do not want to put off the discussion until then.
> We will respond to the other remarks in detail in a separate comment as soon as possible.

---

> ### Author Response · Authors · 2025-11-20
> **Complete Answer - Overview**
>
> We thank the reviewer for raising these vital questions. Before treating each of them in detail, we first provide an overview of our overall answers.
> 1. **Q:** Is IWP more robust than OP for variations in optimization hyperparameters? **A:** Yes, IWP is **considerably more robust**, especially for the choice of optimizer and GroupSum temperature (e.g., **12-13% better absolute accuracy** on SGD and Adadelta).
>
>
> 2. **Q:** Are the generalization benefits of IWP particular to CIFAR? **A:** No, IWPs outperform OP across datasets, as shown in the results we provided in the general comment.
>
>
> 3. **Q:** Do deep IWP DLGNs consistently exhibit lower discretization errors and stable gradients across
>
>
>     3a. … different rounding strategies? **A:** In fact, the neural gate output estimators in the IWP allow us to sustain low discretization errors for *any* rounding threshold, while the OP does not grant this design freedom.
>
>
>     3b. … different encodings? **A:** We show affirmative results for several other encodings (graycode, binary number, interval indicator).
>
>
>     3c. Do IWP DLGNs rely on residual initializations to satisfy the above properties? **A:** No, AND-OR initializations as proposed in the paper achieve similar gradient stability and performance.
>
>
>
>
> 4. **Q:** When increasing the arity of differentiable logic gates,
>
>     4a. … do IWP DLGNs remain computationally feasible (as opposed to the OP) regarding memory requirements and runtime? **A:** Yes, we verified this by profiling CUDA kernels for higher arity.
>
>
>     4b. … do IWP DLGNs preserve both analytically and numerically stable gradients? **A:** They do, thanks to a tree-like accumulation of weighted outputs.
>
>     4c. … can IWP DLGNs still be optimized efficiently? **A:** In fact, IWP DLGNs with arity $n=6$ **converge 8.4x faster** than the IWP DLGNs with arity $n=2$.
>
>
>
>
> ### Experiment Setup
> For all of the following experiments, we consider the same CIFAR-100 model and configuration as in the paper, with a depth scale of 3, i.e., a DLGN with 12 logic layers and 128,000 nodes per layer. We repeat the experiments for 2 seeds and report the mean and standard deviation of the accuracy.

---

> ### Author Response · Authors · 2025-11-20
> **1 - Hyperparameter Variations**
>
> ## 1 - Hyperparameter Variations
> We agree with the reviewer that the original DLGNs are quite sensitive to certain optimization hyperparameters. Besides the choice of optimizer and learning rate, the softmax temperature $\tau$ of the `GroupSum` layer was already identified as a highly sensitive parameter in Petersen et al. (2022, 2024).
> However, we find that IWP DLGNs are far more robust than OP DLGNs against variations of these hyperparameters.
>
>
> ### 1a - Choice of optimizer
> Instead of using the default Adam optimizer as in Petersen et al. (2022), we evaluate the models on three different popular optimizers, i.e.
> - Stochastic Gradient Descent (SGD),
> - Nesterov Accelerated Gradient (NAG),
> - Adadelta
>
>
> For the three new other optimizers, we pick the same learning rate `LR=100.0`.
>
>
> While the IWP DLGN maintains a stable, discretized test accuracy of >30% for all of them, the test accuracy of the OP DLGN deteriorates drastically–OP with Adadelta gets <18% accuracy.
>
>
> | Optimizer  | IWP DLGN    | OP DLGN      |
> | ---------- | ----------- | ------------ |
> | Adam (base)| 32.46 ±0.08 | 31.02 ± 0.09 |
> | NAG        | 30.20 ±0.01 | 27.14 ± 0.21 |
> | SGD        | 30.94 ±0.13 | 18.30 ± 0.05 |
> | Adadelta   | 30.77 ±0.07 | 17.85 ± 0.05 |
>
>
> ### 1b - Learning rate
> With a standard sigmoidal gate output estimator, the IWP DLGNs stably outperform OP DLGNs across the range of learning rates as well (using the Adam optimizer).
>
>
> | Learning Rate | IWP DLGN    | OP DLGN      |
> | ------------- | ----------- | ------------ |
> | 0.16          | 28.83 ±0.08 | 27.10 ± 0.04 |
> | 0.04          | 31.35 ±0.04 | 29.97 ± 0.07 |
> | 0.01 (base)   | 32.46 ±0.08 | 31.02 ± 0.09 |
> | 0.0025        | 30.94 ±0.10 | 30.79 ± 0.20 |
> | 0.000625      | 24.75 ±0.13 | 22.45 ± 0.05 |
>
>
> ### 1c - GroupSum temperature
> Finally, the large discretization errors become particularly apparent when changing the GroupSum temperature $\tau$.
>
> | GroupSum Temperature | IWP DLGN (cont.) | IWP DLGN (disc.) | OP DLGN (cont.) | OP DLGN (disc.) |
> | -------------------- | ---------------- | ---------------- | --------------- | --------------- |
> |   3                  |    26.53 ±0.05   |    18.64 ±0.01   |   26.21 ±0.20   |   12.48 ± 0.05  |
> |  10                  |    29.13 ±0.12   |    27.05 ±0.12   |   28.02 ±0.04   |   25.49 ± 0.11  |
> |  30 (base)           |    32.33 ±0.25   |    32.46 ±0.08   |   30.99 ±0.04   |   31.02 ± 0.09  |
> | 100                  |    24.75 ±0.01   |    24.81 ±0.02   |   21.52 ±0.12   |   21.52 ± 0.13  |
> | 300                  |    18.25 ±0.12   |    18.25 ±0.14   |   14.85 ±0.04   |   14.84 ± 0.03  |

---

> ### Author Response · Authors · 2025-11-20
> **2 - Dataset Variations**
>
> ### 2 - Dataset Variations
> IWP DLGNs consistently outperform OP DLGNs as depth increases, both for simple datasets like MNIST and the more challenging datasets like ImageNet32.
> Since multiple reviewers showed interest in these experiments, we kindly refer the reviewer to the general comment above, where we present the performance results in detail.
> Moreover, we do not expect IWP DLGNs to leverage the spatial structure in vision datasets any better than OP DLGNs, as the connection topology for both is fully randomized and neglects the structure in both spatial and channel dimensions.
> The comparisons to CNNs - which can take advantage of this structure in contrast to the standard version of DLGNs - in Figure 11 and the general comment above substantiate this shortcoming.

---

> ### Author Response · Authors · 2025-11-20
> **3 - Stability against Different Encodings and Roundings**
>
> ## 3 - Stability against Different Encodings and Roundings
> ### 3a - Rounding Strategies
> In fact, we expect IWP DLGNs to exhibit low discretization errors for *any* possible rounding strategy, for the following reason:
> The class of all possible roundings of gate functions can be parametrized by a rounding threshold $t\in[0,1]$ on each gate output, where the default rounding-to-nearest corresponds to $t=0.5$.
> IWP DLGNs can be tailored to any such rounding threshold $t$ by shifting the “activation center” of the neural gate output estimator - i.e., the location $x$ where the non-linear slope is steepest - to this value $t$.
> That way, the neuron can already mimic the rounding behaviour during training.
> The OP falls short of such a feature, as the non-linearity is not applied to the outputs themselves but to the mixture of Boolean functions.
>
>
> ### 3b - Different Encodings
> We evaluate DLGNs on a diverse range of encodings:
> - a binary number system encoding, where 1s express 2-moduli at various granularity
> - a graycode encoding, where 1s express how centered the value is at various subintervals
> - a sparse interval indicator encoding, where a 1 is placed only for the interval in which the value is contained
>
>
> IWP DLGNs exhibit superior performance across all of these encodings.
> | Input encoding      | Num. bits | IWP DLGN    | OP DLGN      |
> | ------------------- | --------- | ----------- | ------------ |
> |  Thermometer (base) |     3     | 32.46 ±0.08 | 31.02 ± 0.09 |
> |  Graycode           |     3     | 32.38 ±0.05 | 31.25 ± 0.05 |
> |  Binary Number      |     3     | 31.57 ±0.05 | 28.90        |
> |  Interval Indicator |     4     | 30.60       | 29.52        |
>
>
> As we have argued for vision datasets, we do not believe that the randomized connection topology allows for exploiting the structure of particular encodings, and that the superior convergence, stability, and discretization arise from the redundancy-free parametrization, as we argued in Section 3 of the paper.
>
>
> ### 3c - Reliance on residual initializations
> Residual initializations are only one of several possible gate initializations that
> - preserve stable gradients, and
> - optimize and converge efficiently.
>
>
> For example, the AND-OR initializations that we have introduced in the paper achieve similar performance.
> | Gate initialization | Train (cont.) | Test (disc.) |
> | ------------------- | ------------- | ------------ |
> | RI (base)           | 46.10 ±1.04   | 32.46 ±0.08  |
> | AND-OR              | 45.40 ±0.10   | 31.66 ±0.11  |
>
>
> For a detailed comparison of the intermediate feature distribution and how these initializations of logic gates affect their optimization dynamics, please see Appendix E of our paper.

---

> ### Author Response · Authors · 2025-11-20
> **4 - Generalization to higher gate arities**
>
> ## 4 - Generalization to higher gate arities
> ### 4a - Computational feasibility
> We have implemented and profiled kernels for up to $n=6$ inputs, which is the largest number of inputs that contemporary lookup-tables in FPGAs admit.
> Even for double-precision floating point computations and fully unrolled single scalar assignment instructions, the register usage of the forward and backward kernels for gate arity 6 remains below the maximum 255 registers per thread on modern NVIDIA GPUs.
> By contrast, OP DLGNs exceed this boundary for gate arity 3 already, where the number of parameters alone becomes 256.
>
>
> Still, the number of registers is evidently larger than for arity 2 kernels, and we do need to decrease the number of threads per thread block by a factor of up to 4 for the backward kernels to meet the register constraints of an NVIDIA streaming multiprocessor.
> Moreover, the computation graph of each logic gate deepens by a factor of 3.
> The overall runtime per training step thus increased by a factor of ~8.
>
>
> ### 4b - Analytical and numerical gradient stability
> Numerical instability is effectively avoided by a tree-like accumulation order in the forward and backward kernels.
> The following illustration showcases the accumulation structure for n=3 inputs.
> ```
> Input Probabilities               Output estimators     Partial Sums 1   Partial Sums 2    Total Sum (Output)
> ------------------------------------------------------------------------------------------------------------
> (1-p1) * (1-p2)  * (1-p3)              w_{000}   ┐    +
> (1-p1) * (1-p2)  *    p3               w_{001}   ┘──────── w_{00}   ┐   +
> (1-p1) *    p2   * (1-p3)              w_{010}   ┐         w_{01}   ┘────── w_{0}   ┐    +
> (1-p1) *    p2   *    p3               w_{011}   ┘                                  |──────── y
>    p1  * (1-p2)  * (1-p3)              w_{100}   ┐         w_{10}   ┐       w_{1}   ┘
>    p1  * (1-p2)  *    p3               w_{101}   ┘         w_{11}   ┘
>    p1  *    p2   * (1-p3)              w_{110}   ┐
>    p1  *    p2   *    p3               w_{111}   ┘
> ```
>
>
> This tree structure is not only a natural choice to minimize instruction dependencies and latency.
> It also ensures that terms have a similar magnitude when added together.
> For example, no matter how extreme $p_1$ and $p_2$ concentrate towards 0 and 1, the “nearest-neighbour” accumulation first accumulates weights whose probability term only differs in $p_3,(1-p_3)$.
> That way, the risk of precision loss due to numerical instability increases only moderately with higher arity.
> We can corroborate this with measurements on the average gradient norm during backpropagation for RI and AND-OR initializations for arity $n=6$ at various layer depths.
>
>
> | Layer | RI (arity 2)  | RI (arity 6) | AND-OR (arity 6) |
> | ----- | ------------- | ------------ | ---------------- |
> | 40    |     2e-5      |     2e-6     |     4e-6         |
> | 20    |     1e-6      |     6e-8     |     3e-9         |
> |  0    |     6e-7      |     3e-8     |     2e-9         |
>
>
> On the contrary, the OP computes a softmax-mixture of Boolean functions, and the softmax weights
> 1. will mostly vanish for larger arities n,
> 2. do not exhibit an invariant structure, such as the probability weights in the IWP. Henceforth, a numerically stable accumulation must depend on the data at runtime and would come at a considerable loss in runtime performance.
>
>
> ### 4c - Optimization efficiency
> Indeed, our IWP DLGNs with arity 6 converge>8x times faster in terms of accuracy matched training steps than the arity 2 counterpart.
>
>
> | Optimization metrics   | IWP RI (arity 2) | IWP RI (arity 6) |
> | --------------------   | ---------------- | ---------------- |
> | Training Loss          |    2.35 ±0.15    |     1.52 ±0.14   |
> | Train Accuracy (cont.) |   46.10 ±1.04    |    73.50         |
> | Test  Accuracy (disc.) |   32.46 ±0.08    |    33.87         |
>
>
> While the arity 2 DLGN reaches its best test accuracy after 210000 training steps, the arity 6 DLGNs has already surpassed this accuracy after 25000 training steps.
> This benefit in convergence speed remains on par with the longer runtime of each training step. However, the current kernel is just a naive implementation without common optimizations such as pipelining or improved memory access patterns that should improve runtime performance.
>
>
> Note that the arity 6 model is larger, as the two models train the same number of LUTs, but arity 2 trains LUT2s while arity 6 trains LUT6s.

---

> ### Author Response · Authors · 2025-11-20
> **Complete Answer - Conclusion**
>
> ### Conclusion
> To conclude:
> - IWP DLGNs are far less brittle to variations of optimization hyperparameters, such as temperature or optimizer
> - IWP DLGNs consistently outperform OP DLGNs across different datasets, encodings, and hyperparameters
> - With IWP DLGNs, higher gate arities are computationally feasible, numerically stable, yield additional convergence speedups, and further improve test accuracy.
>
>
> We hence believe that the reparametrization of DLGNs finally sets the stage to investigate and advance this nascent architecture at a larger scale, and study how it can eventually approach competitive performance to standard neural architectures.
>
>
> Please let us know if there are any further comments or inputs; we are happy to continue the discussion. If you are content with the improvements of our submission, we would appreciate it if you could reflect this in your score.

---

> > ### Comment · Reviewer_Nt7w · 2025-11-27
> >
> > Thank you for submitting your revised paper. I appreciate the effort you've put into addressing the comments and improving the results. The updated work demonstrates significant progress, and I am impressed by the improvements. Congratulations on your efforts. Based on this revision, I am revising my rating to an 8. Keep up the great work, and I look forward to seeing how this research evolves.

---

### Official Review · Reviewer_H3Lt · 2025-11-04

**Soundness:** 3
**Presentation:** 3
**Contribution:** 3
**Rating:** 8
**Confidence:** 3

**Summary:**

This paper proposes an input-wise parameterization (IWP) of logic gate networks with tailored initializations that mitigate vanishing gradient issues of  DLGNs when increasing depth. Specifically, the paper points out that in previous DLGN, since there are negation pairs of logic gate in each layer, initializing the weight of each gate independently will cause gradient cancellation during back propagation. To address this issue, the paper reparameterize the logic function with fewer independent components. Experimental results show that the proposed method achieve better performance than existing DLGNs when increasing network depth.

**Strengths:**

1. The paper is well written and easy to follow
2. The paper is well motivated, as addressing the vanishing gradient issue of deep DLGNs will improve their applicability for more challenging tasks
3. The analysis of the root cause of vanishing gradient issue in DLGN , and the solution proposed by the paper make intuitive sense and are all backed by solid proof.
4. Experimental results and ablation study are comprehensive and positively support the proposed design.

**Weaknesses:**

1. The paper conducts experiment only on CIFAR 100.

**Questions:**

1. It would be better if the paper add comparison experiment on more complex dataset like ImageNet 32

---

> ### Author Response · Authors · 2025-11-15
> **Experiments on more datasets in general comment above**
>
> We thank the reviewer for their thoughtful and constructive remarks. Since multiple reviewers asked for experiments on other datasets than CIFAR-100, we have posted a general comment above where we present preliminary results for ImageNet32, CIFAR-10, MNIST and Fashion-MNIST. We are conducting more experiments, but we do not want to put off the discussion until then.
> We will respond to the other remarks in detail in a separate comment as soon as possible.

---

> ### Author Response · Authors · 2025-11-20
> **Complete Answer**
>
> We thank the reviewer for encouraging us to overcome this limitation.
> We have now conducted more experiments, and these are our findings:
> - The IWP DLGN **consistently outperforms** the OP on various datasets, from MNIST to ImageNet32. We kindly refer the reviewer to the general comment above for details.
> - IWP DLGNs are **far more robust** than OP DLGNs against variations of hyperparameters, such as the GroupSum softmax temperature or the choice of optimizer (e.g., **12-13% better absolute accuracy** on SGD and Adadelta). For detailed experiments, please see Section 1 in our reply to Reviewer Nt7w.
> - Increasing the gate arity finally becomes feasible, which yields **another >8x convergence speedup** in training steps for gate arity 6. For details, please see Section 4 of our reply to Reviewer Nt7w.

---

### Official Review · Reviewer_fw6a · 2025-11-04

**Soundness:** 3
**Presentation:** 3
**Contribution:** 3
**Rating:** 6
**Confidence:** 2

**Summary:**

The paper tackles the problem of vanishing gradients, discretization errors, and high training cost of Differentiable logic gate networks (DLGNs). The authors claimed to root cause the issues and proposed a reparametrization solution to resolve it. Redundant parameters of input grates are replaced while maintaining the representability. With binary inputs, it can achieve 4x smaller model size, 1.86x backward pass speedup and 8.5x fewer training steps to converge.

**Strengths:**

1. Less weight parameters compared w/ the original DLGN paper.

**Weaknesses:**

DLGN seems to have a low test accuracy, which makes it less appealing as a practically useful solution.

**Questions:**

It makes sense that light weight gate parameters help resolving the vanishing gradient issue. How is this  solution working on a larger data set like imagenet?

---

> ### Author Response · Authors · 2025-11-15
> **Experiments on more datasets in general comment above**
>
> We thank the reviewer for their thoughtful and constructive remarks. Since multiple reviewers asked for experiments on other datasets than CIFAR-100, we have posted a general comment above where we present preliminary results for ImageNet32, CIFAR-10, MNIST and Fashion-MNIST. We are conducting more experiments, but we do not want to put off the discussion until then.
> We will respond to the other remarks in detail in a separate comment as soon as possible.

---

> ### Author Response · Authors · 2025-11-20
> **Complete Answer**
>
> We thank the reviewer for raising these concerns, and approach them individually in the following.
>
>
> ### 1 - ImageNet and hyperparameter variation: Consistently superior, far more robust
> We have now conducted more experiments, and these are our findings:
> - The IWP DLGN **consistently outperforms** the OP on various datasets, from MNIST to ImageNet32. We kindly refer the reviewer to the general comment above for details.
> - IWP DLGNs are **far more robust** than OP DLGNs against variations of hyperparameters, such as the GroupSum softmax temperature or the choice of optimizer (e.g., **12-13% better absolute accuracy** on SGD and Adadelta). For detailed experiments, please see Section 1 in our reply to Reviewer Nt7w.
>
>
> ### 2 - Low test accuracy of DLGNs
> We emphasize that vanilla CNNs do not perform much better than DLGNs when provided with the same quantized input data.  We demonstrate this both for ImageNet32 in the general comment above and for CIFAR-100 in Figure 11 in our paper. The encoding, hence, acts as an information bottleneck that curbs the overall achievable representation power.
>
>
> Apart from that, the reparametrization of DLGNs alleviates major bottlenecks that prevented the original architecture from scaling along multiple dimensions. We want to stress two beneficial aspects of this scalability:
> - Increasing the gate arity finally becomes feasible, which yields **another >8x convergence speedup** in training steps for gate arity 6. For details, please see Section 4 of our reply to Reviewer Nt7w.
> - For the deeper, more recent convolutional DLGN architecture from Petersen et al. (2024), we also remark that increasing depth exposes a considerably larger performance gap than for the original architecture.
>
>
> Thanks to this robust and efficient refinement of the underlying parametrization, the remaining hurdles towards competitive performance can now be investigated at a larger scale.
> The reparametrization is surely not the only, but an important first milestone on this quest.
>
>
> For example, the randomized connection topology prevents DLGNs from leveraging structure along both the spatial and channel dimensions. Figure 11 in Appendix B.5 demonstrates the latter, while the performance increase of convolutional DLGNs in Petersen et al. (2024) confirms the former.
>
> Please let us know if there are any further comments or inputs; we are happy to continue the discussion. If you are content with the improvements of our submission, we would appreciate it if you could reflect this in your score.

---

### Author Response · Authors · 2025-11-15
**Performance results on other datasets**

We thank the reviewers for their thoughtful remarks. We will respond to each of them in detail later.
To not put off the discussion until then, we already want to provide some preliminary performance results on other datasets than CIFAR-100, since multiple reviewers asked for that.
Additional experiments were conducted on the RGB datasets ImageNet32 and CIFAR-10, and the grayscale datasets MNIST and Fashion-MNIST.
For the RGB datasets, we use the baseline CIFAR-10 M LGN architecture from Petersen et al. (2022) with 4 layers of 128k neurons each, just as we have done for CIFAR-100.

For the RGB datasets we use the following baseline model:
```
Sequential(
  (0): Encoding(thermometer, resolution=3)
  (1): LogicLayerIWP(9216, 128000)
  (2): LogicLayerIWP(128000, 128000)
  (3): LogicLayerIWP(128000, 128000)
  (4): LogicLayerIWP(128000, 128000)
  (5): GroupSum(k=10, tau=...)
)
```
For CIFAR-10 we set tau to 100 as was done by Petersen et al. (2022), and to accommodate the 100-fold increase of classes in ImageNet32, we adopt the heuristic that was proposed in Petersen et al. (2022) and shrink the softmax temperature by sqrt(100).


For the simpler gray-scale datasets, we use 1-threshold thermometer encoding and a smaller width of 32k neurons to again align with Petersen et al. (2022) resulting in the model below.
```
Sequential(
  (0): Encoding(thermometer, resolution=1)
  (1): LogicLayerIWP(784, 32000)
  (2): LogicLayerIWP(32000, 32000)
  (3): LogicLayerIWP(32000, 32000)
  (4): LogicLayerIWP(32000, 32000)
  (5): GroupSum(k=10, tau=100.0)
)
```
All of the networks were trained for 250k steps. Due to time constraints, the results are not yet averaged over multiple seeds.

We have gathered the test accuracies of the discretized logic gate networks (in %) into the following table, along with the performance of a vanilla two-layer CNN. The latter is included to give a baseline performance to expect with the encoding in place. Note that, as shown in Figure 11 in the paper, the CNN is very sensitive to the number of thresholds.

| Test accuracy (disc.) | ImageNet32 | Cifar-100 | Cifar-10 | Fashion-MNIST | MNIST |
|--------|------------|-----------|----------|---------------|-------|
|DLGN OP | 3.48*      |   27.2    |   55.9   |     80.5      |  91.0 |
|DLGN IWP| 3.53*      |   29.3    |   56.8   |     81.5      |  93.3 |
|CNN     | 5.00       |   20.6    |   58.2   |     75.7      |  90.3 |

*These numbers were obtained for models with twice as many layers as the baseline models.


For ImageNet32, the following tables depict the continuous training and discretized test accuracies of the IWP DLGN across depth scales 1 to 5.
We observe a monotonic improvement in both train and test accuracies for deeper models, even with the same number of training iterations. However, the improvement from depth scale 4 to 5 is small.

| ImageNet32 - Train accuracy (cont.) | DLGN IWP |
|-------------------------------------|----------|
| depth scale 1                       |   5.37   |
| depth scale 2                       |   6.84   |
| depth scale 3                       |   7.62   |
| depth scale 4                       |   8.98   |
| depth scale 5                       |   9.38   |

| ImageNet32 - Test accuracy (disc.) | DLGN IWP |
|------------------------------------|----------|
| depth scale 1                      |   2.90   |
| depth scale 2                      |   3.53   |
| depth scale 3                      |   3.59   |
| depth scale 4                      |   3.81   |
| depth scale 5                      |   3.82   |

When comparing the IWP to the OP, we do observe that IWP marginally outperforms the OP as depth increases. This aligns with our hypothesis that IWP is better suited to optimize deep DLGNs.

| ImageNet32 - Test accuracy | DLGN OP  | DLGN IWP |
|----------------------------|----------|----------|
| depth scale 1              |   3.05   |   2.90   |
| depth scale 2              |   3.48   |   3.53   |
| depth scale 3              |   3.48   |   3.59   |
| depth scale 4              |   3.48   |   3.81   |
| depth scale 5              |   3.56   |   3.82   |

We will conduct more experiments over the following days and include these in the paper.

---

> ### Author Response · Authors · 2025-11-18
> **Update: ImageNet32 test accuracy - Measurements for all depth scales now available**
>
> In the comment above, we have updated the table that compares the test accuracy of the OP and IWP on ImageNet32 across depth scales. Now, measurements for all depth scales 1 to 5 are included. Again, these measurements are not averaged across multiple seeds yet. We will notify you once we have included exhaustive experiment results in the paper.

---

### Author Response · Authors · 2025-11-24
**WMT’14 Language Modeling Experiments**

We have added experiments on the WMT’14 dataset to evaluate IWP DLGNs beyond image classification tasks. For these experiments, we used the implementation of a language model as described in [1]. Table 1 in the updated PDF (also included below) shows that IWP consistently outperforms the original OP architecture on WMT’14, demonstrating that the reparameterization improves both performance and robustness across vision and language tasks.

| Model     | ImageNet32      | CIFAR-100       | CIFAR-10         | Fashion-MNIST    | MNIST            | WMT'14    |
|-----------|----------------|----------------|-----------------|-----------------|-----------------|-----------|
| DLGN OP   | 4.84 ± 0.02    | 27.7 ± 0.05    | 55.33 ± 0.23    | 81.39 ± 0.07    | 92.43 ± 0.17    | 15.11 ± 0.64 |
| DLGN IWP  | 4.93 ± 0.02    | 29.5 ± 0.02    | 57.47 ± 0.20    | 82.34 ± 0.15    | 94.02 ± 0.08    | 17.38 ± 0.04 |
| CNN       | 5.19 ± 0.37    | 39.2 ± 0.07    | 64.01 ± 0.16    | 77.66 ± 0.85    | 92.91 ± 1.61    | --          |


Reference:
[1] Simon Buhrer, Andreas Plesner, Till Aczel, and Roger Wattenhofer. Recurrent deep differentiable logic gate networks, 2025. URL: https://arxiv.org/abs/2508.06097

---

### Author Response · Authors · 2025-12-03

Given the changes to the rebuttal, we wanted to provide a summary of the main discussion points.

## Results on new datasets
Multiple reviewers asked for results on more datasets, so we ran IWP, OP, and the baseline CNN on ImageNet32, CIFAR-10, Fashion-MNIST, and MNIST. We also ran IWP and OP on the English to German WMT’14 translation task (here we compare the BLEU scores). The results (mean ± std) can be seen below, where we see that IWP is always better than OP.

| Model | ImageNet32 | CIFAR-100 | CIFAR-10 | Fashion-MNIST | MNIST | WMT'14    |
|---|---|---|---|---|---|---|
| DLGN OP | 4.84 ± 0.02    | 27.7 ± 0.05 | 55.33 ± 0.23 | 81.39 ± 0.07 | 92.43 ± 0.17    | 15.11 ± 0.64 |
| DLGN IWP | 4.93 ± 0.02 | 29.5 ± 0.02 | 57.47 ± 0.20 | 82.34 ± 0.15 | 94.02 ± 0.08    | 17.38 ± 0.04 |
| CNN | 5.19 ± 0.37 | 39.2 ± 0.07 | 64.01 ± 0.16 | 77.66 ± 0.85 | 92.91 ± 1.61 | -- |
* These numbers are for models with a depth scale of 2.

## Higher arity results
Since our parameterization scales better in terms of the number of inputs, we also tested models with 6 inputs (arity 6). The results for IWP can be seen below for CIFAR-100 and a depth scale of 3. See ‘Experiment Setup’ in the reply to Reviewer Nt7w for details (https://openreview.net/forum?id=EaGQ5luZtf&noteId=8Y43akM1JU). With the higher arity, the model converges to a higher accuracy. Further modern FPGAs have LUTs with up to six inputs, so we can effectively train models that directly fit the underlying hardware.

|Accuracy| Arity 2 | Arity 6 |
| --- | --- | --- |
| Train (cont.) | 46.10±1.04 | 73.50 |
| Test (disc.) | 32.46±0.08 | 33.87 |

## How to read the results considering what SOTA ViT and CNN models can do?
For a detailed discussion of this and the references, we refer to our comments addressing the questions Reviewer Pghb raised during the rebuttal. We provide a summary below.

DLGNs are currently focused on working well in low-resource settings. However, they also allow for ultra-fast inference, so they can also work well in domains where low latency is essential.

At the moment, DLGNs achieve lower accuracy compared to modern CNNs, but CNNs have benefited from over two decades of intensive research, engineering, and community-wide innovation. In contrast, DLGNs are a very new class of models, still in their infancy, so, unsurprisingly, there is currently a performance gap. We are not claiming that DLGNs will necessarily reach the level of CNNs. However, we expect that gap to shrink as the field of differentiable logic networks matures.
To appreciate how early-stage models behave, it is useful to recall how CNNs performed before the major breakthroughs of the past fifteen years. A convolutional model on CIFAR-10 would typically achieve only around seventy to eighty percent test accuracy [2], far below what even modest architectures achieve today. Only through a series of foundational ideas did CNNs become scalable and competitive.
Furthermore, we would like to highlight this statement from He et al. (2016) [6]: “When deeper networks are able to start converging, a degradation problem has been exposed: with the network depth increasing, accuracy gets saturated (which might be unsurprising) and then degrades rapidly. Unexpectedly, such degradation is not caused by overfitting, and adding more layers to a suitably deep model leads to higher training error.”

Thus, given this, we believe that DLGNs will be able to scale to much better performance. While many techniques, such as convolution, can be used to train better DLGNs, there is still a lot of work to figure out how to benefit from modern regularization methods or design new methods for DLGNs. We showed in Figure 28 that the straightforward applications of dropout to DLGNs did not lead to better results.

---

### Meta-Review · Area_Chair_F9xu · 2026-01-07

**Summary:**

This paper proposes a new parameterization technique for differentiable logic gate networks. The reviewers found that this manuscript is easy to follow and successfully addresses the vanishing gradient problem in differentiable logic gate networks. I agree with these evaluations and recommend the acceptance of this paper.

**Reviewer Concerns:**

The authors thoroughly address all reviewers' concerns.

**Reviewer Scores:**

During the discussion period, Reviewers Pghb and Nt7w increase their scores from 4, 6 to 6, 8, respectively. I do not expect other reviewers to edit their scores.

---

### Decision · Program_Chairs · 2026-01-26

Accept (Poster)